# Rethinking gradient sparsification
# as total error minimization

**Atal Narayan Sahu**
KAUST

**Aritra Dutta**
KAUST

**Ahmed M. Abdelmoniem**
KAUST

**Trambak Banerjee**
University of Kansas

**Marco Canini**
KAUST

**Panos Kalnis**
KAUST

## Abstract

Gradient compression is a widely-established remedy to tackle the communication bottleneck in distributed training of large deep neural networks (DNNs). Under the error-feedback framework, Top-$k$ sparsification, sometimes with $k$ as little as $0.1\%$ of the gradient size, enables training to the same model quality as the uncompressed case for a similar iteration count. From the optimization perspective, we find that Top-$k$ is the communication-optimal sparsifier given a per-iteration $k$ element budget. We argue that to further the benefits of gradient sparsification, especially for DNNs, a different perspective is necessary — one that moves from per-iteration optimality to consider optimality for the entire training.

We identify that the *total error* — the sum of the compression errors for all iterations — encapsulates sparsification throughout training. Then, we propose a communication complexity model that minimizes the total error under a communication budget for the entire training. We find that the *hard-threshold sparsifier*, a variant of the Top-$k$ sparsifier with $k$ determined by a constant hard-threshold, is the optimal sparsifier for this model. Motivated by this, we provide convex and non-convex convergence analyses for the hard-threshold sparsifier with error-feedback. We show that hard-threshold has the same asymptotic convergence and linear speedup property as SGD in both the case, and unlike with Top-$k$ sparsifier, has no impact due to data-heterogeneity. Our diverse experiments on various DNNs and a logistic regression model demonstrate that the hard-threshold sparsifier is more communication-efficient than Top-$k$. Code is available at `https://github.com/sands-lab/rethinking-sparsification`.

## 1 Introduction

With the emergence of huge DNNs consisting of hundreds of millions to billions of parameters [12, 50], distributed data-parallel training [66] is an increasingly important workload. As the training process typically spans several compute nodes (or workers) that periodically exchange the local gradient vectors at each iteration of the optimizer (e.g., SGD), communication among nodes remains in many cases the main performance bottleneck [32, 40, 46].

Lossy gradient compression techniques are becoming a common approach to rein in communication efficiency [62]. In particular, sparsification, which sends only a subset of gradient coordinates (e.g., Top-$k$ [4, 8] sends the $k$ largest gradient coordinates by magnitude in each iteration), may significantly reduce data volumes and thus speed up training. However, due to its lossy nature, compression raises a complex trade-off between training performance and accuracy. For instance, Agarwal et al. [3] note that training ResNet-18 on CIFAR-100 using sparsification speeds up training significantly ($3.6\times$), but it also degrades final accuracy by 1.5%. On the other hand, Lin et al. [37] reports a $500\times$ data

35th Conference on Neural Information Processing Systems (NeurIPS 2021).

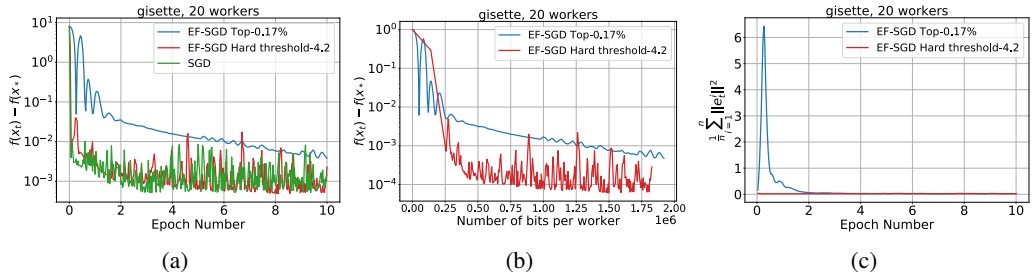

Figure 1: Convergence of Top-$k$ and Hard-threshold for a logistic regression model on `gisette` LIBSVM dataset with 20 workers: (a) Functional suboptimality vs. epochs; (b) functional suboptimality vs. bits communicated; (c) error norm vs. epochs. Hard-threshold converges as fast as the baseline, no compression SGD and much faster than Top-$k$ because of a smaller total-error than Top-$k$.

reduction via sparsification under deep gradient compression (DGC) for ResNet-50 on ImageNet while preserving the same final accuracy when adopting a carefully-tuned warmup phase.

The vast literature on gradient compression largely considers a fixed communication budget per iteration while leaving it up to practitioners to grapple with specifying an additional hyper-parameter that determines the degree of compression before training begins. Meanwhile, recent adaptive Top-$k$ sparsifiers [3, 65] empirically demonstrate that tuning the degree of compression in different phases of DNN training yields a more communication-efficient scheme than a fixed compression scheme (e.g., a static $k$ for Top-$k$). However, these works lack a theoretical framework proving that adaptive compression enjoys better convergence guarantees than the fixed compression scheme.

This raises a fundamental question: *Given a fixed communication budget, is there a provably better communication scheme than fixed per-iteration compressed communication?* We first observe that Top-$k$ is the communication-optimal sparsifier for a fixed per-iteration communication budget (§4.3). Then, our insight is that by adopting a different perspective that accounts for *the effect of sparsification throughout training*, a more efficient communication scheme is possible under a revised notion of optimality that considers an overall communication budget (instead of a per-iteration budget).

We consider sparsification by using the *error-feedback* (EF) mechanism [8, 53], a delayed gradient component update strategy that is instrumental for the convergence of the state-of-the-art sparsifiers [10]. Let $e_t$ denote the error arising due to sparsification at iteration $t$. In EF, this error is added to the gradient update at iteration $t + 1$. We identify that the term affecting the non-convex convergence in EF-SGD is the *total-error:* $\sum_t \|e_t\|^2$ [33, 54].

Directly minimizing the total-error is not possible; thus, Top-$k$ minimizes $\|e_t\|^2$ at each iteration. We argue that it is possible to focus on the *sum of* $\|e_t\|^2$ and devise a communication scheme that achieves a smaller total-error than any fixed communication sparsifier. We demonstrate that to achieve this change of perspective; it is sufficient to consider a practical yet straightforward mechanism that is a natural counterpart of Top-$k$: the *hard-threshold sparsifier*, which communicates the gradient coordinates with magnitude greater than or equal to a fixed given threshold, $\lambda \geq 0$, in each iteration. Although the two sparsifiers are in an equivalence relation (a given $\lambda$ corresponds to a $k$), under the total-error minimization perspective, we adopt a *fixed* threshold, $\lambda$, which implies a *variable $k$* at every iteration.

To illustrate intuitively why this change of perspective yields benefits, consider the following example. Figure 1 shows an experiment in the distributed setting where 20 workers train a 6,000-parameter logistic regression model on the `gisette` LIBSVM dataset [14] by using the Top-$k$ and hard-threshold sparsifiers, configured to send the *same data volume*.[1] The loss function is strongly convex and has a unique minimizer, $x^\star$, therefore, a unique optimum, $f(x^\star)$. We see that hard-threshold converges at the same speed as SGD while communicating $\sim 600\times$ less data, whereas Top-$k$ has a significantly slower convergence speed. We attribute this to the fact that Top-$k$ has a large error accumulation in the initial 500 iterations, while the error magnitude for hard-threshold is less than 0.04 throughout training (cf. Figure 1c). Our results with DNN training also reflect this insight (§6).

---

[1]We train for 10 epochs and set $k = 0.17\%$ for Top-$k$, and $\lambda = 4.2$ for hard-threshold.

Moreover, the hard-threshold sparsifier has computational benefits over Top-$k$ sparsifier, as hard-threshold's underlying filtering operation requires $d$ comparisons in each iteration, where $d$ is the number of parameters. In contrast, Top-$k$ is a compute-intensive sparsifier (e.g., on CPU, the computational complexity is $\mathcal{O}(d \log_2 k)$ [48]). For GPUs, several optimized implementations are proposed but they rely on the data distribution and are efficient only for a small $k$ [48]. For instance, PyTorch uses Radix select algorithm [5] which has a computational complexity of $\mathcal{O}\left(\lceil b/r \rceil \, d\right)$ where $b$ is the number of bits to represent gradient values and $r$ is the radix size [42].

Finally, while the hard-threshold sparsifier already exists in the literature [20, 55], we are the first to formally study it and theoretically demonstrate its benefits as an adaptive counterpart of Top-$k$. Moreover, our argument in favor of hard-threshold precisely falsifies the claim by Dryden et al. [18] that stopped its widespread adoption — *a hard-threshold may lead to a degenerate situation when the EF in gradient compression builds up.*

This paper makes the following contributions:

**Communication complexity model (§4).** We propose a communication complexity model that captures the effects of compression in the entire optimization process. We allow for variable communication in each iteration by only imposing a total communication budget. We show that the hard-threshold sparsifier is the communication-optimal sparsifier in this model.

**Absolute compressors (§5).** We identify that the hard-threshold sparsifier, along with other existing compressors [16, 46], belongs to the class of *absolute compressors*, which have an absolute bound on the error. Absolute compressors have not been formally studied before with EF. We show that absolute compressors with EF converge for both strongly convex and non-convex loss functions. In both cases, similar to the $\delta$-contraction operators [33], absolute compressors enjoy the same asymptotic convergence with linear speedup (with respect to the number of workers) as no-compression SGD. However, $\delta$-contraction operators have a worse dependence on $\delta$ in the distributed setting with heterogeneous data, while absolute compressors do not have such an anomaly.

**Experiments (§6).** We conduct diverse experiments on both strongly convex and non-convex (for DNNs) loss functions to substantiate our claims. Our DNN experiments include computer vision, language modeling, and recommendation tasks, and our strongly convex experiment is on logistic regression. We find that the hard-threshold sparsifier is consistently more communication-efficient than the Top-$k$ sparsifier given the same communication budget.

## 2   Related work

**Gradient compression** techniques can be broadly classified into quantization [7, 16, 33, 47, 60], sparsification [4, 37, 59], hybrid compressors [9, 18, 55], and low-rank methods [57, 58]. The state-of-the-art compressors are biased $\delta$-contraction operators [37, 57], see §4.5. We refer to [62] for a recent survey and quantitative evaluation of these techniques.

**Error-feedback** (EF) or memory was first empirically used in [47, 55]. However, [33, 53, 54] were the first to give a convergence analysis of the EF framework, which was extended to the distributed setup in [10, 64]. Recently, [61] proposed *error-reset*, a different form of EF, while [29] introduced another alternative by communicating a compressed version of the error. EF has also been combined with variance-reduction [22, 43] and acceleration [44].

**Communication-optimal compression.** [6, 15, 21, 45] devised a communication-optimal compressor by minimizing the worst-case compression factor[2] under a per-vector communication budget.

**Adaptive compression.** [59] designed an adaptive sparsifier that minimizes expected sparsity of the compressed vector under a given variance budget. While AdaQS [23] periodically doubles the quantization states in QSGD [7] to reduce the compression factor, DQSGD [63] sets the number of quantization states proportional to the gradient norm. ACCORDION [3] chooses a low compression ratio if training is in a critical regime [2], and a high compression ratio otherwise.

---

[2]For a vector $x$ and a possibly randomized compression operator $\mathcal{C}$, we denote the compression error as $\mathbb{E}_{\mathcal{C}}[\|x - \mathcal{C}(x)\|^2]$, and compression factor as $\mathbb{E}_{\mathcal{C}}[\|x - \mathcal{C}(x)\|^2]/\|x\|^2$.

**Efficient Top-$k$ estimation** is a focus of many recent works. While [37] estimates the Top-$k$ threshold on a randomly sampled subset, [1, 49] estimate it by fitting parametric statistical distributions. [25] estimates the threshold every $1,000$ iterations. These works determine a threshold to approximately determine a *fixed* Top-$k$ set, while we use a hard-threshold to determine the $k$ in each iteration.

## 3    Background on Error-Feedback (EF) SGD

Consider the following distributed optimization problem with $n$ workers:
$$\min_{x\in\mathbb{R}^d} f(x) := \tfrac{1}{n}\sum_{i\in[n]} f_i(x), \tag{1}$$

where $f_i(x) = \mathbb{E}_{z_i\sim\mathcal{D}_i} l(x; z_i)$ denotes the loss function evaluated on input $z_i$ sampled from $\mathcal{D}_i$, the data-distribution at the $i^{th}$ worker. Let $g_{i,t}$ denote the stochastic gradient computed at $i^{th}$ worker at iteration $t$ such that $g_{i,t} = \nabla f_i(x_t) + \xi_{i,t}$ with $\mathbb{E}[\xi_{i,t}|x_t] = \mathbf{0}$.

Algorithm 1 gives the pseudo-code of compressed EF SGD [8, 33, 53] to solve (1). Let $e_{i,t}$ denote the locally accumulated error at $i^{th}$ worker due to compression from previous steps. Adding this error to the current gradient, $\gamma_t g_{i,t}$ provides the corrected update, $p_{i,t}$, where $\gamma_t > 0$ is the step-size. This corrected update is further compressed and exchanged with other machines, and the local error is updated for the next step.

---

**Algorithm 1:** Distributed EF SGD

---

**Input:** $\mathcal{C}$-compressor
Initialize $x_0 \in \mathbb{R}^d$;
**for** *worker* $i \in [n]$ **in parallel do**
    Initialize error $e_{i,0} = 0$;
    **for** *iteration* $t = 0, \ldots, T - 1$ **do**
        $p_{i,t} \leftarrow e_{i,t} + \gamma_t g_{i,t}$          /* Incorporate EF into update */;
        $\Delta_{i,t} \leftarrow \gamma_t \mathcal{C}(\frac{p_{i,t}}{\gamma_t})$ ;
        $e_{i,t+1} \leftarrow p_{i,t} - \Delta_{i,t}$          /* Update error */;
        $\bar{\Delta}_t = \frac{1}{n}\sum_{i\in[n]}\Delta_{i,t}$      /* Exchange and average $\Delta_{i,t}$ among workers */;
        $x_{t+1} \leftarrow x_t - \bar{\Delta}_t$;

---

*Remark* 1. In Algorithm 1, we have $\Delta_{i,t} \leftarrow \gamma_t \mathcal{C}(\frac{p_{i,t}}{\gamma_t})$, while [8, 33, 53] consider $\Delta_{i,t} \leftarrow \mathcal{C}(p_{i,t})$. We do this to extend the EF framework for absolute compressors. We note that Algorithm 1 is more general as $\gamma_t \mathcal{C}(\frac{p_{i,t}}{\gamma_t})$ is equivalent to $\mathcal{C}(p_{i,t})$ for all known $\delta$-contraction operators.

### 3.1    Assumptions

We consider the following general assumptions on the loss function.

**Assumption 1.** *(Smoothness) The function, $f_i : \mathbb{R}^d \to \mathbb{R}$ at each worker, $i \in [n]$ is L-smooth, i.e., for every $x, y \in \mathbb{R}^d$ we have, $f_i(y) \le f_i(x) + \langle \nabla f_i(x), y - x \rangle + \frac{L}{2}\|y - x\|^2$.*

**Assumption 2.** *(Global minimum) There exists $x^\star$ such that, $f(x^\star) = f^\star \le f(x)$, for all $x \in \mathbb{R}^d$.*

**Assumption 3.** *($(M, \sigma^2)$ bounded noise) [54] For every stochastic noise $\xi_{i,t}$, there exist $M, \sigma^2 \ge 0$, such that $\mathbb{E}[\|\xi_{i,t}\|^2 \mid x_t] \le M\|\nabla f_i(x_t)\|^2 + \sigma^2$, for all $x_t \in \mathbb{R}^d$.*

*Remark* 2. The above assumption implies $\mathbb{E}[\|g_{i,t}\|^2 \mid x_t] \le (M + 1)\|\nabla f_i(x_t)\|^2 + \sigma^2$. This general noise model does not uniformly bound the second moment of stochastic gradients as in [9, 33, 64].

**Assumption 4.** *($(C, \zeta^2)$ bounded similarity) The variance of gradients among workers is bounded, i.e., there exist constants, $C, \zeta \ge 0$ such that, $\frac{1}{n}\sum_{i\in[n]}\|\nabla f_i(x) - \nabla f(x)\|^2 \le C\|\nabla f(x)\|^2 + \zeta^2$, for all $x \in \mathbb{R}^d$.*

*Remark* 3. If all the workers have the same training data, all $f_i$ are the same, resulting in $C, \zeta = 0$. This assumption is an extension from [31, 36], which consider $C = 0$.

For the *convergence of strongly convex functions*, we require an additional assumption as follows.

**Assumption 5.** *($\mu$-strong convexity) The functions, $f_i : \mathbb{R}^d \to \mathbb{R}$ are $\mu$-strongly convex, i.e., there exists $\mu \ge 0$, such that $f_i(y) \ge f_i(x) + \langle \nabla f_i(x), y - x \rangle + \frac{\mu}{2}\|x - y\|^2$, for all $x \in \mathbb{R}^d$.*

**Convergence of EF-SGD.** The following result shows the convergence of EF-SGD [33] in minimizing general smooth functions for a single worker ($n = 1$) case.

**Theorem 1.** *[33, 54] Let Assumption 1, 2 and 3 hold. Then Algorithm 1 with a constant step-size, $\gamma$ where $\gamma \leq \frac{1}{2L(M+1)}$ and $n = 1$ follows*

$$\frac{1}{T} \sum_{t=0}^{T-1} \mathbb{E}\|\nabla f(x_t)\|^2 \leq \frac{4(f(x_0) - f^\star)}{\gamma T} + 2\gamma L\sigma^2 + 2L^2 \sum_{t=0}^{T-1} \frac{\gamma^2 \mathbb{E}\|g_t + \frac{e_t}{\gamma} - \mathcal{C}(g_t + \frac{e_t}{\gamma})\|^2}{T}.$$

*Remark* 4. Theorem 1 is a simplified version of the distributed case for $n > 1$ and quoted to emphasize the effect of compression between the error-corrected gradient, $g_t + \frac{e_t}{\gamma}$, and its compressed form, $\mathcal{C}(g_t + \frac{e_t}{\gamma})$. The term that *solely accounts for the effect of compression in the entire training process* is the *total-error*: $\sum_{t=0}^{T-1} \mathbb{E}\|e_{t+1}\|^2 = \sum_{t=0}^{T-1} \gamma^2 \mathbb{E}\|g_t + \frac{e_t}{\gamma} - \mathcal{C}(g_t + \frac{e_t}{\gamma})\|^2$.

# 4 A communication complexity perspective to sparsification

We now propose a communication complexity model and contrast it with the existing communication-optimal strategies. We note that our use of the word *communication-optimal* is in the sense of optimization upper-bounds. Convergence analyses of compressed SGD capture the effect of compression via the compression error, and this effect is always inverse — the lower the compression error, the better the optimization upper bound [22, 29]. Therefore, ours and the existing works [6, 15, 21, 45] design *communication-optimal* strategies by optimizing the compression error related term. We start with a sparse approximation problem that we encounter in our subsequent discussions.

## 4.1 A sparse approximation problem

Let $p \in \mathbb{R}^m$ be a given vector. We want to approximate $p$ with a sparse vector, $q$, that has at most $0 < \tau \leq m$ non-zero elements. Formally, we write the *constrained sparse approximation* problem as:

$$q^\star = \arg\min_{q \in \mathbb{R}^m} \|p - q\|^2 \quad \text{subject to } \|q\|_0 \leq \tau, \tag{2}$$

where $\|\cdot\|_0$ denotes the number of non-zero elements in a vector. Problem (2) and its variants are well studied and arise in signal processing [13, 17, 19] and matrix approximation [11, 51].

**Lemma 1.** *The solution $q^\star$ to (2) is obtained by keeping Top-$\tau$ magnitude entries from $p$ and setting the rest to zeros.*

## 4.2 Minimizing the total-error is not possible

Let $\mathfrak{C}$ denote the class of all compressors. We constrain to the class of deterministic sparsifiers, denoted by $\mathbf{S} \subset \mathfrak{C}$, but one can similarly consider other subclasses in $\mathfrak{C}$. For each $x \in \mathbb{R}^d$, a deterministic sparsifier, $\mathcal{C}_p$ with sparsification parameter, $p$ determines a sparse support set, $S_p(x) \subseteq [d]$ and sparsifies as

$$\mathcal{C}_p(x) = \sum_{i \in S_p(x)} x[i] e_i,$$

where $e_i$ denotes the $i^{th}$ standard basis in $\mathbb{R}^d$, and $x[i]$ denotes the corresponding element in $x$. For example, for hard-threshold sparsifier, $\mathcal{C}_\lambda$, we have $S_\lambda(x) = \{i \mid |x[i]| \geq \lambda\}$. Motivated by Theorem 1 and Remark 4, we now propose the following communication complexity model:

$$\min_{\mathcal{C} \in \mathbf{S}} \sum_{t=0}^{T-1} \mathbb{E}\|g_t + \frac{e_t}{\gamma} - \mathcal{C}(g_t + \frac{e_t}{\gamma})\|^2 \quad \text{subject to } \sum_{t=0}^{T-1} \|\mathcal{C}(g_t + \frac{e_t}{\gamma})\|_0 \leq K, \tag{3}$$

where $K$ is the budget on the number of elements communicated in $T$ iterations. However, solving (3) is intractable owing to complex DNN loss functions and multiple sources of randomness.

## 4.3 Top-$k$ is communication-optimal for a per-iteration $k$ element budget

To simplify (3), one can focus individually at the error at each iteration. Based on this, we show in Lemma 2 that Top-$k$ has the best compression error among all sparsifiers under a per-iteration $k$-element communication budget.

**Lemma 2.** *Given the gradient $g_t$ and error $e_t$ at iteration $t$, Top-$k$ sparsifier achieves the optimal objective for the optimization problem:*

$$\min_{\mathcal{C} \in \mathbf{S}} \|g_t + \frac{e_t}{\gamma} - \mathcal{C}(g_t + \frac{e_t}{\gamma})\|^2 \quad \textit{subject to } \|\mathcal{C}(g_t + \frac{e_t}{\gamma})\|_0 \leq k. \tag{4}$$

Similar to Lemma 1, (4) is solved when $\mathcal{C}(g_t + \frac{e_t}{\gamma})$ contains the $k$ highest magnitude elements of $g_t + \frac{e_t}{\gamma}$. That is, when $\mathcal{C}$ is the Top-$k$ sparsifier. Additionally, based on the above model, a per-iteration $k$-element communication budget (resulting in a total budget of $kT$ elements throughout training), implies that Top-$k$ is performed at each iteration. However, to have a more communication-efficient compression, we require a communication complexity model that (*i*) better captures total-error in Theorem 1; and (*ii*) allows for adaptive communication, i.e., sends variable data in each iteration.

### 4.4 A communication complexity model for adaptive sparsification

Although the total-error cannot be minimized (§4.2), Lemma 2 motivates us to consider a *simplified model* that can capture the total-error. Instead of $(g_t + \frac{e_t}{\gamma})_{t=0}^{T-1}$, we consider a fixed sequence $(a_t)_{t=0}^{T-1}$ and examine the following communication complexity model:

$$\min_{\mathcal{C} \in \mathbf{S}} \sum_{t=0}^{T-1} \|a_t - \mathcal{C}(a_t)\|^2 \quad \textit{subject to } \sum_{t=0}^{T-1} \|\mathcal{C}(a_t)\|_0 \leq K, \tag{5}$$

where $K \in \mathbb{N}$ denotes the total communication budget. For the sake of simplicity, we assume that no two elements in $(a_t)_{t=0}^{T-1}$ have the same magnitude.

Let $\mathcal{A} \in \mathbb{R}^{dT}$ be formed by stacking $(a_t)_{t=0}^{T-1}$ vertically and consider the following sparse approximation problem:

$$\min_{B \in \mathbb{R}^{dT}} \|\mathcal{A} - B\|^2 \quad \textit{subject to } \|B\|_0 \leq K, \tag{6}$$

Note that (6) allows for all $B$ that are formed by stacking $(\mathcal{C}(a_t))_{t=0}^{T-1}$ vertically, for some sparsifier $\mathcal{C}$ satisfying $\sum_{t=0}^{T-1} \|\mathcal{C}(a_t)\|_0 \leq K$. Therefore, the optimal objective for (6) is a lower bound to the optimal objective for (5). Let $\mathcal{A}_{(i)}$ denote the element with $i^{th}$ largest magnitude in $\mathcal{A}$, and since no two elements have the same magnitude, we have, $\mathcal{A}_{(i+1)} \neq \mathcal{A}_{(i)}$, for all $i \in [dT]$. Then, $B = \mathcal{C}_\lambda(A)$ with $\lambda \in (\mathcal{A}_{(K+1)}, \mathcal{A}_{(K)}]$ contains the Top-$K$ magnitude entries from $A$, and therefore, by Lemma 1 is optimal for (6). Moreover, since hard-threshold is an element-wise sparsifier, $\mathcal{C}_\lambda(A)$ is equivalent to stacking $(\mathcal{C}_\lambda(a_t))_{t=0}^{T-1}$ vertically. Therefore, $\mathcal{C}_\lambda$ with $\lambda = (\mathcal{A}_{(K+1)}, \mathcal{A}_{(K)}]$ achieves optimal objective in (5). The following lemma formalizes this.

**Lemma 3.** *$\mathcal{C}_\lambda$ is optimal for the communication complexity model (5). That is, for every budget $K$, there exists a $\lambda \geq 0$ such that $\mathcal{C}_\lambda$ minimizes (5).*

### 4.5 Discussion

To capture the effect of compression, existing works [7, 33, 53] use a bound on the *compression factor*, $\max_{x \in \mathbb{R}^d} \frac{\mathbb{E}_{\mathcal{C}} \|\mathcal{C}(x) - x\|^2}{\|x\|^2}$. We formally define them as relative compressors.

**Definition 2. Relative Compressor** [7, 53]. An operator, $\mathcal{C} : \mathbb{R}^d \to \mathbb{R}^d$ is a *relative compressor* if for all vector, $x \in \mathbb{R}^d$ it satisfies

$$\mathbb{E}_{\mathcal{C}} \|x - \mathcal{C}(x)\|^2 \leq \Omega \|x\|^2, \tag{7}$$

where $\Omega > 0$ is the compression factor and the expectation, $\mathbb{E}_{\mathcal{C}}$, is taken with respect to the randomness of $\mathcal{C}$. $\delta$-*contraction operators* [33, 53] with $\Omega = 1 - \delta$ and $\delta \in (0, 1]$, are special cases of relative compressors.

Top-$k$ is a $\delta$-contraction operator with $\delta = \frac{k}{d}$ [53]. Therefore, by (7), Top-$k$ allows for larger compression error with larger inputs. Our communication complexity model demonstrates that this might not necessarily be a good idea. Moreover, with EF, a large error at any iteration has a cascading effect — a large $e_t$ results in a large $\gamma g_t + e_t$, out of which only $k/d$ fraction of the total components are kept by the Top-$k$ strategy. This results in a large $e_{t+1}$ (see §C.2). Figure 1 shows that this *error-buildup* has severe implications on the total-error. On the other hand, the hard-threshold performs a variable Top-$k$ in each iteration and sends an element as soon as its magnitude is bigger than the threshold. This prohibits the error build-up.

**Comparison with existing communication-optimal compression strategies.** Since the compression factor, $\Omega$ solely determines the effect of compression in convergence [29, 54], many recent works [6, 15, 21, 45] propose communication-optimal compression strategies by optimizing for $\Omega$ under a *communication budget* for each vector, i.e., they propose to solve

$$\min_{\mathcal{C} \in \mathfrak{C}} \max_{x \in \mathbb{R}^d} \frac{\mathbb{E}_{\mathcal{C}} \|x - \mathcal{C}(x)\|^2}{\|x\|^2} \quad \text{subject to } \mathtt{Bits}(\mathcal{C}(x)) \le B, \tag{8}$$

where $\mathtt{Bits}(\mathcal{C}(x))$ denotes the number of bits needed to encode $\mathcal{C}(x)$. We stress that while the compression affected term in Theorem 1 has the sum of compression errors over the iterations, the above communication complexity model only captures the compression factor.

# 5 Absolute compressors and their convergence

Motivated by the previous section, we formally define *absolute compressors* — compressors that have an absolute bound on the error.

**Definition 3. Absolute Compressor.** An operator, $\mathcal{C} : \mathbb{R}^d \to \mathbb{R}^d$ is an *absolute compressor* if there exists a $\upsilon > 0$ such that for all vectors, $x \in \mathbb{R}^d$ it satisfies

$$\mathbb{E}_{\mathcal{C}} \|x - \mathcal{C}(x)\|^2 \le \upsilon^2. \tag{9}$$

In contrast to the relative compressors in (7), the compression error (or variance) of absolute compressors is bounded by a constant, independent of $x$. Based on the above definition, hard-threshold is an absolute sparsifier with $\upsilon^2 = d\lambda^2$. The stochastic rounding schemes, used for model quantization, with bounded rounding error in [24] are absolute compressors. Precisely, any rounding scheme, with rounding error bounded by $\epsilon$, is an absolute compressor with $\upsilon^2 = d\epsilon^2$. Similarly, the scaled integer rounding scheme in [46] is an absolute compressor. While this class of compressors existed in the literature, we are the first to provide their convergence result with an EF.

## 5.1 Convergence results

Inspired by [54], we establish convergence of EF-SGD (Algorithm 1) with absolute compressors. Convergence analysis for the momentum case [64] can be extended similarly. However, we do not include it for brevity, and the existing analyses do not show any benefit over vanilla SGD. Similarly, analysis for error-reset [61] and local updates [9, 61] can also be extended. We provide convergence results for the convex and non-convex cases and compare them to $\delta$-contraction operators. We start with a bound on the error for absolute compressors.

*Remark* 5. **(Error bound)** For all $i \in [n], t \in \{0, \dots T - 1\}$, we have

$$\mathbb{E}_{\mathcal{C}}[\|e_{i,t+1}\|^2 \mid p_{i,t}] = \mathbb{E}_{\mathcal{C}}\|p_{i,t} - \gamma_t \mathcal{C}(\tfrac{p_{i,t}}{\gamma_t})\|^2 = \gamma_t^2 \mathbb{E}_{\mathcal{C}}\|\tfrac{p_{i,t}}{\gamma_t} - \mathcal{C}(\tfrac{p_{i,t}}{\gamma_t})\|^2 \le \gamma_t^2 \upsilon^2.$$

A similar absolute bound for $\delta$-contraction operators requires the bounded gradient assumption [33, 9], but absolute compressors achieve this by design.

### 5.1.1 Convex convergence

Let $\bar{x}_T = \frac{1}{W_T} \sum_{t=0}^{T} w_t x_t$ be the weighted average of the iterates with weights, $w_t \ge 0$ and $W_T = \sum_{t=0}^{T} w_t$. Additionally, let $P_t := \mathbb{E}[f(\bar{x}_t)] - f^\star$ be the expected suboptimality gap at the average iterate. Further denote $R_t := \|x_t - x^\star\|^2$ and $D := \frac{1}{n} \sum_{i=1}^{n} \|\nabla f_i(x^\star)\|^2$. With these notations, we quote the strongly convex ($\mu > 0$) and convex ($\mu = 0$) convergence results for absolute compressors, and compare them with the $\delta$-contraction operators from [10] for distributed case ($n \ge 1$). The results below are for specific choices of step-sizes and weights; we refer to §B.4.1 for these choices.

**Theorem 4.** *Let $\mu > 0$ and Assumptions 1, 2, 3, and 5 hold. Then the iterates, $\{x_t\}_{t \ge 0}$ of Algorithm 1 with an absolute compressor, $\mathcal{C}_\upsilon$, a constant step-size, $\gamma(T)$ with $\gamma(T) \le \frac{1}{4L(1+2M/n)}$ follow* [3]

$$P_T = \tilde{\mathcal{O}} \left( L R_0 (1 + M/n) \exp\left[ -\frac{\mu T}{8L(1+2M/n)} \right] + \frac{\sigma^2 + MD}{\mu n T} + \frac{L\upsilon^2}{\mu^2 T^2} \right).$$

---

[3]The $\tilde{\mathcal{O}}$ notation hides constants and factors polylogarithmic in the problem parameters.

*Remark* 6. Under the same setting as in Theorem 4, iterates of Algorithm 1 with $\delta$-contraction operators follow:

$$P_T = \tilde{\mathcal{O}}\left( LR_0 \left(\frac{\sqrt{1+M\delta}}{\delta}\right) \exp\left[-\frac{\mu\delta T}{16\sqrt{3}L\sqrt{2+M\delta}}\right] + \frac{\sigma^2+MD}{\mu nT} + \frac{L(D(1+M\delta)+\delta\sigma^2)}{\mu^2\delta^2 T^2}\right).$$

Remark 6 implies, in distributed settings with heterogeneous data ($D \neq 0$), $\delta$-contraction operators have an $1/\delta^2$ dependence on $\delta$, as compared to an $1/\delta$ dependence in the homogeneous case ($D = 0$). In contrast, absolute compressors have the same $v^2$ dependence on $v$ in both cases. Therefore, we conjecture it is beneficial to use absolute compressors in settings such as federated learning [38], where data heterogeneity is widely encountered.

**Theorem 5.** *Let $\mu = 0$ and Assumptions 1, 2, 3, and 5 hold with $D \neq 0$. Then the iterates, $\{x_t\}_{t\geq 0}$ of Algo. 1 with an absolute compressor, $\mathcal{C}_v$, a constant step-size, $\gamma(T)$ with $\gamma(T) \leq \frac{1}{4L(1+2M/n)}$ follow*

$$P_T = \mathcal{O}\left( \frac{\sqrt{(\sigma^2+MD)R_0}}{\sqrt{nT}} + \frac{\left(\frac{nLv^2}{\sigma^2+MD}+L(1+M/n)\right)R_0}{T}\right).$$

*Remark* 7. Theorem 5 holds when both $\sigma^2$ and $D$ are not simultaneously zero. Typically, we encounter heterogeneous data settings where $D \neq 0$, and Theorem 5 holds. In case both $\sigma^2$ and $D$ are zero, we get $\mathcal{O}(\frac{(LvR_0)^{\frac{2}{3}}}{T^{\frac{2}{3}}} + \frac{L(1+M/n))R_0}{T})$ convergence.

*Remark* 8. Under the same setting as in Theorem 5, iterates of Algorithm 1 with $\delta$-contraction operators follow:

$$P_T = \mathcal{O}\left( \frac{\sqrt{(\sigma^2+MD)R_0}}{\sqrt{nT}} + \frac{\left(\frac{L\sqrt{1+M\delta}}{\delta} + \frac{nL(D(1+M\delta)+\delta\sigma^2)}{\delta^2(\sigma^2+MD)}\right)R_0}{T}\right).$$

Similar to Remark 6, we observe that $\delta$-contraction operators have $1/\delta^2$ dependence on $\delta$ in the heterogeneous case, and a $1/\delta$ dependence in the homogeneous case, while absolute compressors have no such anomaly.

Designing a variance-reduced algorithm [22, 43, 44] by using absolute compressors with EF is a fruitful direction of future research.

### 5.1.2 Non-convex convergence

**Theorem 6.** *(Non-convex convergence of absolute compressors) Let Assumptions 1, 2, 3, and 4 hold. Then the iterates, $\{x_t\}_{t\geq 0}$ of Algorithm 1 with an absolute compressor and a constant step-size $\gamma \leq \frac{n}{2L(M(C+1)+n)}$ follow*

$$\frac{1}{T}\sum_{t=0}^{T-1} \mathbb{E}\|\nabla f(x_t)\|^2 \leq \frac{4(f(x_0)-f^\star)}{\gamma T} + \frac{2\gamma L(M\zeta^2+\sigma^2)}{n} + 2\gamma^2 L^2 v^2.$$

Alongside, we compare with the non-convex convergence for $\delta$-contraction operators in a distributed setting. The existing analyses tackle this by using a stronger *uniform bounded gradient* assumption [64, 33, 20]. We use weaker Assumption 3 and 4, to establish the convergence analysis.

**Theorem 7.** *(Non-convex convergence of $\delta$-contraction operators) Let Assumptions 1, 2, 3, and 4 hold. Then the iterates, $\{x_t\}_{t\geq 0}$ of Algorithm 1 with a $\delta$-compressor and a constant step-size $\gamma \leq \min\{\frac{n}{2L(M(C+1)+n)}, \frac{1}{2L(2/\delta+M)\sqrt{C+1}}\}$ follow*

$$\frac{1}{T}\sum_{t=0}^{T-1} \mathbb{E}\|\nabla f(x_t)\|^2 \leq \frac{8(f(x_0)-f^\star)}{\gamma T} + \frac{4\gamma L(M\zeta^2+\sigma^2)}{n} + \frac{8\gamma^2 L^2}{\delta}\left(\left(\frac{2}{\delta}+M\right)\zeta^2 + \sigma^2\right).$$

Again, similar to Remarks 6 and 8, for $\delta$-contraction operators, we find the $1/\delta^2$ (heterogeneous case, $\zeta \neq 0$) vs. $1/\delta$ (homogeneous case, $\zeta = 0$) anomaly, while absolute compressors have $v^2$ dependence on $v$ in both homogeneous and heterogeneous cases.

*Remark* 9. With appropriate choices of step-size, both absolute compressors and $\delta$-contraction operators with EF-SGD achieve the same $\mathcal{O}(1/\sqrt{nT})$ asymptotic rate of SGD. See Corollary 1 in §B.3 for the full result.

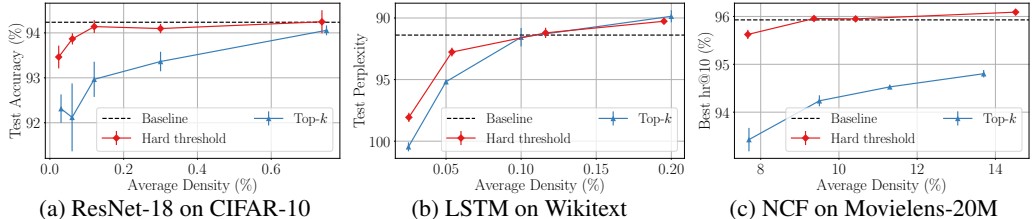

(a) ResNet-18 on CIFAR-10     (b) LSTM on Wikitext     (c) NCF on Movielens-20M

Figure 2: **Test metric vs. Data volume.** For 3 benchmarks, average test quality with std. dev. over 3 runs. The dashed black line denotes the no compression baseline.

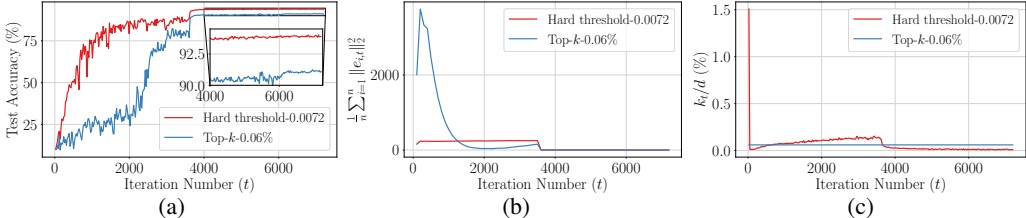

(a)        (b)        (c)

Figure 3: **Convergence of Top-$k$ and Hard-threshold for ResNet-18 on CIFAR-10 at $0.06\%$ average density:** (a) Test-accuracy vs. Iterations, (b) Error-norm vs. Iterations, (c) Density ($k_t/d$) vs. Iterations. $k = 0.06\%$ of $d$, and $\lambda = 0.0072$. Hard-threshold has better convergence than Top-$k$ because of a smaller total-error.

## 6 Experiments

**Experimental setup.** We compare Top-$k$ and hard-threshold sparsifiers on image classification, language modelling, and recommendation tasks. We use different optimizers: vanilla SGD, SGD with Nesterov momentum, and ADAM [34]. All experiments were run on an 8-GPU cluster, using `Allgather` as the communication primitive. We perform compression in the standard layer-wise fashion [20, 37, 47] and follow the EF strategy used in [57]. For hyper-parameter configuration, comparison with entire-model compression, discussion on different EF approaches, experiments without EF, and experiments with logistic regression, we refer to Appendix C.

**Test metric vs. Data volume.** We tune the sparsification parameters for both sparsifiers such that they send similar total data volumes during training. We use *average density*: $\frac{1}{T}\sum_{t=0}^{T-1}\frac{k_t}{d}$ as a measure of total data volume, where $k_t$ denotes the number of elements transmitted in iteration $t$. Figure 2 shows the average test quality across three repetitions with different initial random seeds. We observe that fixing the average density, *hard-threshold consistently has better test performance* than Top-$k$. For ResNet-18 on CIFAR-10, we observe that hard-threshold at an average density of $0.12\%$ almost achieves the baseline accuracy and is better than Top-$k$ at $0.75\%$ density ($\sim 6\times$ more total data volume). For LSTM on Wikitext, at an average density of $0.025\%$, hard-threshold has $> 2$ better perplexity than Top-$k$. For NCF on Movielens-20M, hard-threshold has $> 1\%$ better Hit-Rate@10 at all considered average densities.

We now demonstrate that hard-threshold has faster convergence because of a smaller total-error in comparison to Top-$k$. In Figure 3, we introspect a run with average density of $0.06\%$ from Figure 2a. In Figure 3a, while hard-threshold converges to an accuracy of $93.9\%$, Top-$k$ achieves $91.1\%$ accuracy. At the same time, in Figure 3b, we observe large error-accumulation in the initial $1,200$ iterations for Top-$k$. Consequently, hard-threshold has a significantly lower total-error than Top-$k$, and therefore has better convergence. This observation about large error accumulation for Top-$k$ is consistent across all our benchmarks (see §C.2).

**Comparison against ACCORDION.** We compare against the state-of-the-art adaptive sparsifier: ACCORDION [3] on CIFAR-10 and CIFAR-100 datasets. ACCORDION shifts between two user-defined $k$ values: $k_{\max}$ and $k_{\min}$, by using Top-$k_{\max}$ when the training is in a *critical regime*, else using Top-$k_{\min}$. We compare against ACCORDION with hard-threshold $\lambda = \frac{1}{2\sqrt{k_{\min}}}$. For complete experiment details; see §C.4.

We report the CIFAR-10 result in Table 1, while the CIFAR-100 result is reported in §C.4. Each setting is repeated with 6 different seeds and we report the average. For the CIFAR-10 dataset,

Table 1: Comparison against ACCORDION [3] on CIFAR-10.

| Network | Method | Accuracy (%) | Average Density (%) |
|---------|--------|--------------|---------------------|
| ResNet-18 | Top-1% ($k_{\max}/d$) | 94.1 | 1.00 ($1\times$) |
| | Top-0.1% ($k_{\min}/d$) | 93.2 | 0.10 ($10\times$) |
| | ACCORDION | 93.5 | 0.53 ($1.9\times$) |
| | Hard-threshold ($\frac{1}{2\sqrt{k_{\min}}}$) | **94.0** | **0.13 (7.7$\times$)** |
| GoogleNet | Top-1% ($k_{\max}/d$) | 94.1 | 1.00 ($1\times$) |
| | Top-0.1% ($k_{\min}/d$) | 92.9 | 0.10 ($10\times$) |
| | ACCORDION | 93.4 | 0.47 ($2.1\times$) |
| | Hard-threshold ($\frac{1}{2\sqrt{k_{\min}}}$) | **94.2** | **0.13 (7.7$\times$)** |
| SENet18 | Top-1% ($k_{\max}/d$) | 94.0 | 1.00 ($1\times$) |
| | Top-0.1% ($k_{\min}/d$) | 92.5 | 0.10 ($10\times$) |
| | ACCORDION | 93.5 | 0.47 ($2.1\times$) |
| | Hard-threshold ($\frac{1}{2\sqrt{k_{\min}}}$) | **94.2** | **0.14 (7.1$\times$)** |

we observe that hard-threshold has $0.5\% - 0.8\%$ higher test accuracy than ACCORDION and is approximately $3.5\times$ more communication efficient than ACCORDION. For the CIFAR-100 dataset, except the ResNet-18 model, we observe that hard-threshold obtains more than $0.8\%$ higher accuracy than ACCORDION with more than $1.26\times$ communication savings over ACCORDION.

**How to tune the hard-threshold?** We use the non-convex convergence results from §5.1.2 to suggest a hard-threshold value which has better convergence than Top-$k$ with parameter $k$ for non-convex loss functions (including DNNs). Let $\hat{M}$, $\hat{\zeta}$, and $\hat{\sigma}$ be the estimates of $M$, $\zeta$, and $\sigma$, respectively, in Assumptions 3 and 4. We set the threshold as $\lambda \sim \frac{2}{\sqrt{k}}\sqrt{\left(\frac{2d}{k} + \hat{M}\right)\hat{\zeta}^2 + \hat{\sigma}^2}$; see discussion in §D. The $\lambda$ in Table 1 is derived from simplifying this formula. But how to tune the hard-threshold such that it achieves no-compression baseline performance with the least total-data transmission remains an open question. We remark, as of now, this question remains unanswered for Top-$k$ as well.

**When and when not to use hard-threshold?** In a standard cluster setting with a dedicated network, the speedup in terms of per-iteration training time due to gradient compression depends on the characteristics of the DNN being trained [46]. One of the determining characteristics is the extent to which the communication phase overlaps with computation. If the fraction of non-overlapped communication is significant, then communication is a bottleneck, even if Top-$k$ compression is applied. However, in the case of hard-threshold sparsification (configured for the same total communication volume), during iterations with high data transmission, the non-overlapped communication remains; but during iterations with low data transmission, non-overlapped communication reduces, thereby reducing the overall training time. On the other hand, if there is complete overlap between computation and communication for Top-$k$, then a hard-threshold with the same total communication volume may introduce non-overlapped communication in some iterations with high data transmission, thereby increasing overall training time. Here, we ignored two important aspects of hard-threshold: (i) Hard-threshold may require fewer iterations to a target accuracy owing to its better statistical efficiency, and that (ii) hard-threshold has negligible computation overhead in comparison to Top-$k$.

## 7 Conclusion

We proposed a total-error perspective to compressed communication that captures the effect of compression during the entire training process. Under this, we showed that the hard-threshold sparsifier is more communication-efficient than the state-of-the-art Top-$k$ sparsifier, and is a principled way to perform adaptive sparsification. Absolute compressors – the class of compressors in which hard-threshold belongs – have promising convergence in the heterogeneous data settings, which is a prominent issue in Federated Learning [38]. As the EF framework is also applicable to Local SGD [52], we hope that this inspires more communication-efficient versions of Local SGD that adaptively determine when to communicate, rather than naively communicating in fixed intervals. Furthermore, similar to hard-threshold, we believe adaptive absolute compressor counterparts of quantization schemes and low-rank methods can also be developed.

**Acknowledgements**

We thank Chen-Yu Ho and Hang Xu for many helpful discussions and the reviewers for their feedback. For computer time, this research used the resources of the Supercomputing Laboratory at KAUST.

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
