## Contents

Appendices are supporting material that has not been peer-reviewed.

# A Notations

In this paper, by $[d]$ we denote the set of $d$ natural numbers $\{1, 2, \cdots, d\}$. We denote the $\ell_2$ norm of a vector $x \in \mathbb{R}^d$ by $\|x\|$, and the $\ell_1$ and $\ell_\infty$-norms are denoted by $\|x\|_1$ and $\|x\|_\infty$, respectively. By **0** we denote a vector of all 0s in $\mathbb{R}^d$. In the proofs, we use the notation $\mathbb{E}_t[\cdot]$ to denote expectation conditioned on the iterate, $x_t$, that is, $\mathbb{E}[\cdot|x_t]$.

# B Convergence analysis

In this section, we provide the proofs of convex and non-convex convergence results of the absolute compressors with EF, and compare them with that of the $\delta$-contraction operators, and vanilla SGD.

## B.1 Overview of results

In §B.2, we provide the technical lemmas and inequalities necessary for the analyses. In §B.3 we provide the non-convex convergence results, and §B.4 contains the convex convergence results.

## B.2 Technical results

**Lemma 4.** *If $a, b \in \mathbb{R}^d$ then the Young's inequality is: For all $\rho > 0$, we have*

$$\|a + b\|^2 \leq (1 + \rho)\|a\|^2 + (1 + \rho^{-1})\|b\|^2. \tag{10}$$

*Alternatively,*

$$2 \langle a, b \rangle \leq \rho\|a\|^2 + \rho^{-1}\|b\|^2. \tag{11}$$

**Lemma 5.** *For $a_i \in \mathbb{R}^d$ we have:*

$$\|\frac{1}{n} \sum_{i=1}^{n} a_i\|^2 \leq \frac{1}{n} \sum_{i=1}^{n} \|a_i\|^2. \tag{12}$$

**Lemma 6.** *[54] Let $r_0, c \geq 0$, $d, T > 0$, and $0 < \gamma \leq \frac{1}{d}$. Then choosing $\gamma = \min(\frac{1}{d}, \sqrt{\frac{r_0}{cT}})$, the following holds:*

$$\frac{r_0}{\gamma T} + c\gamma \leq \frac{dr_0}{T} + \frac{2\sqrt{cr_0}}{\sqrt{T}}$$

*Proof.* We consider two cases. If $\frac{r_0}{cT} \leq \frac{1}{d^2}$, then choosing the step-size $\gamma = \left(\frac{r_0}{cT}\right)^{1/2}$, we get

$$\frac{r_0}{\gamma T} + c\gamma \leq \frac{2\sqrt{cr_0}}{\sqrt{T}}.$$

Else, if $\frac{r_0}{cT} > \frac{1}{d^2}$, then choosing $\gamma = \frac{1}{d}$, we get

$$\frac{r_0}{\gamma T} + c\gamma \leq \frac{dr_0}{T} + \frac{c}{d} \leq \frac{dr_0}{T} + \frac{\sqrt{cr_0}}{\sqrt{T}}.$$

Combining both bounds, we get the result. $\qquad\square$

**Lemma 7.** *Let $r_0, b \geq 0$, $c, d, T > 0$, and $0 < \gamma \leq \frac{1}{d}$. Then choosing $\gamma = \min(\frac{1}{d}, \sqrt{\frac{r_0}{cT}})$, the following holds:*

$$\frac{r_0}{\gamma T} + c\gamma + b\gamma^2 \leq \frac{dr_0}{T} + \frac{2\sqrt{cr_0}}{\sqrt{T}} + \frac{br_0}{cT}.$$

*Proof.* The proof follows similar to Lemma 6. We consider two cases. If $\frac{r_0}{cT} \le \frac{1}{d^2}$, then choosing the step-size $\gamma = \left(\frac{r_0}{cT}\right)^{1/2}$, we get

$$\frac{r_0}{\gamma T} + c\gamma + b\gamma^2 \le \frac{2\sqrt{cr_0}}{\sqrt{T}} + \frac{br_0}{cT}.$$

Else, if $\frac{r_0}{cT} > \frac{1}{d^2}$, then choosing $\gamma = \frac{1}{d}$, we get

$$\frac{r_0}{\gamma T} + c\gamma + b\gamma^2 \le \frac{dr_0}{T} + \frac{c}{d} + \frac{b}{d^2} \le \frac{dr_0}{T} + \frac{\sqrt{cr_0}}{\sqrt{T}} + \frac{br_0}{cT}.$$

Combining both bounds, we get the result. $\qquad\square$

**Lemma 8.** *[54] Let $r_0 \ge 0$, $d, T > 0$, and $0 < \gamma \le \frac{1}{d}$. Then choosing $\gamma = \min(\frac{1}{d}, \left(\frac{r_0}{bT}\right)^{1/3})$, the following holds:*

$$\frac{r_0}{\gamma T} + b\gamma^2 \le \frac{dr_0}{T} + \frac{2(br_0)^{2/3}}{T^{2/3}}.$$

*Proof.* We consider two cases. If $\frac{r_0}{bT} \le \frac{1}{d^3}$, then choosing the step-size $\gamma = \left(\frac{r_0}{bT}\right)^{1/3}$, we get

$$\frac{r_0}{\gamma T} + b\gamma^2 \le \frac{2(br_0)^{2/3}}{T^{2/3}}.$$

Else, if $\frac{r_0}{bT} > \frac{1}{d^3}$, then choosing $\gamma = \frac{1}{d}$, we get

$$\frac{r_0}{\gamma T} + b\gamma^2 \le \frac{dr_0}{T} + \frac{b}{d^2} \le \frac{dr_0}{T} + \frac{(br_0)^{2/3}}{T^{2/3}}.$$

Combining both bounds, we get the result. $\qquad\square$

**Lemma 9.** *For every non-negative sequence $\{r_t\}_{t \ge 0}$ and parameters, $a > 0, b, c \ge 0, T \ge 2, \phi \ge 1$, decreasing step-sizes $\{\gamma_t := \frac{2}{a(\phi+t)}\}_{t \ge 0}$, and weights $\{w_t := (\phi + t)\}_{t \ge 0}$, satisfy*

$$\Psi_T := \frac{1}{W_T} \sum_{t=0}^{T} \left( \frac{w_t}{\gamma_t}(1 - a\gamma_t)r_t - \frac{w_t}{\gamma_t}r_{t+1} + c\gamma_t w_t + b\gamma_t^2 w_t \right) \le \frac{4c}{aT} + \frac{a\phi^2 r_0}{T^2} + \frac{16b\ln(T)}{a^2 T^2},$$

*where $W_T := \sum_{t=0}^{T} w_t$.*

*Proof.* This proof is motivated from Lemma 11 in [54]. We observe

$$\frac{w_t}{\gamma_t}(1 - a\gamma_t)r_t = \frac{a}{2}(\phi + t)(\phi + t - 2)r_t = \frac{a}{2}((\phi + t - 1)^2 - 1)r_t \le \frac{a}{2}(\phi + t - 1)^2 r_t. \quad (13)$$

By plugging in the definition of $\gamma_t$ and $w_t$ in $\Psi_t$, we find

$$\Psi_T \overset{(13)}{\le} \frac{1}{W_T} \sum_{t=0}^{T} \left( \frac{a}{2}(\phi + t - 1)^2 r_t - \frac{a}{2}(\phi + t)^2 r_{t+1} \right) + \sum_{t=0}^{T} \frac{2c}{aW_T} + \sum_{t=0}^{T} \frac{4b}{a^2(\phi + t)W_T}$$

$$\le \frac{a(\phi - 1)^2 r_0}{2W_T} + \frac{2c(T + 1)}{aW_T} + \frac{4b}{a^2 W_T} \sum_{t=0}^{T} \frac{1}{\phi + t}.$$

By using $(\phi - 1)^2 \le \phi^2$, $W_T = \sum_{t=0}^{T}(\phi + t) \ge \frac{(2\phi + T)(T+1)}{2} \ge \frac{(T+1)(T+2)}{2}$, and $\sum_{t=0}^{T} \frac{1}{\phi+t} \le \sum_{t=0}^{T} \frac{1}{1+t} \le \ln(T + 1) + 1$, we have

$$\Psi_T \le \frac{a\phi^2 r_0}{(T+1)(T+2)} + \frac{4c}{a(T+2)} + \frac{8b(\ln(T+1) + 1)}{a^2(T+1)(T+2)}.$$

For $T \ge 2$, we have $\frac{(\ln(T+1)+1)}{(T+1)(T+2)} \le \frac{2\ln(T)}{T^2}$. By using this, we get

$$\Psi_T \le \frac{a\phi^2 r_0}{T^2} + \frac{4c}{aT} + \frac{16b\ln(T)}{a^2 T^2}.$$

Hence the result. $\qquad\square$

**Lemma 10.** *(Lemma D.2 in [22]) For every non-negative sequence $\{r_t\}_{t\geq 0}$ and parameters, $d \geq a > 0$, $b, c, T \geq 0$, with a bound on the step-size $\gamma_t \leq \frac{1}{d}$, there exists a constant step-size,*

$$\gamma_t = \gamma = \min\{\frac{1}{d}, \frac{\ln(\max\{2, \min\{a^2 r_0 T^2/c, a^3 r_0 T^3/b\}\})}{aT}\}$$

*and weights, $w_t := (1-a\gamma)^{-(t+1)}$, such that for all $T$ satisfying $\frac{\ln(\max\{2,\min\{a^2 r_0 T^2/c, a^3 r_0 T^3/b\}\})}{T} \leq 1$, we have*

$$\Psi_T := \frac{1}{W_T}\sum_{t=0}^{T}\left(\frac{w_t}{\gamma_t}(1-a\gamma_t)r_t - \frac{w_t}{\gamma_t}r_{t+1} + c\gamma_t w_t + b\gamma_t^2 w_t\right) = \tilde{\mathcal{O}}\left(dr_0\exp\left[-\frac{a}{d}T\right] + \frac{c}{aT} + \frac{b}{a^2 T^2}\right).$$

*Proof.* Substituting the values for $\gamma_t$ and $w_t$, we get

$$\Psi_T = \frac{1}{\gamma W_T}\sum_{t=0}^{T}(w_{t-1}r_t - w_t r_{t+1}) + \frac{c\gamma}{W_T}\sum_{t=0}^{T}w_t + \frac{b\gamma^2}{W_T}\sum_{t=0}^{T}w_t$$

$$\leq \frac{r_0}{\gamma W_T} + c\gamma + b\gamma^2$$

$$\leq \frac{r_0}{\gamma}\exp[-a\gamma T] + c\gamma + b\gamma^2, \tag{14}$$

where we use $W_T \geq w_T \geq (1 - a\gamma)^{-T} \geq \exp[a\gamma T]$ in the last inequality. To tune $\gamma$, we consider following two cases:

- If $\frac{1}{d} \geq \frac{\ln(\max\{2,\min\{a^2 r_0 T^2/c, a^3 r_0 T^3/b\}\})}{aT}$, then we choose $\gamma = \frac{\ln(\max\{2,\min\{a^2 r_0 T^2/c, a^3 r_0 T^3/b\}\})}{aT}$ and (14) becomes $\tilde{\mathcal{O}}(\frac{c}{aT} + \frac{b}{a^2 T^2})$, as
- If $\frac{1}{d} < \frac{\ln(\max\{2,\min\{a^2 r_0 T^2/c, a^3 r_0 T^3/b\}\})}{aT}$, then we choose $\gamma = \frac{1}{d}$ and (14) becomes $\tilde{\mathcal{O}}(dr_0\exp\left[-\frac{a}{d}T\right] + \frac{c}{aT} + \frac{b}{a^2 T^2})$.

Combining both bounds, we get the result. $\qquad\square$

The recurrence relation in the next lemma is instrumental for perturbed iterate analysis of Algorithm 1 used in both convex and non-convex cases.

**Lemma 11.** *Let $\bar{e}_t = \frac{1}{n}\sum_{i=1}^{n}e_{i,t}$, $\bar{g}_t = \frac{1}{n}\sum_{i=1}^{n}g_{i,t}$, and $\bar{p}_t = \frac{1}{n}\sum_{i=1}^{n}p_{i,t}$. Define the sequence of iterates $\{\tilde{x}_t\}_{t\geq 0}$ as $\tilde{x}_t = x_t - \bar{e}_t$, with $\tilde{x}_0 = x_0$. Then $\{\tilde{x}_t\}_{t\geq 0}$ satisfy the recurrence: $\tilde{x}_{t+1} = \tilde{x}_t - \gamma_t \bar{g}_t$.*

*Proof.* We have

$$\tilde{x}_{t+1} = x_{t+1} - \bar{e}_{t+1} = x_t - (\bar{e}_t + \gamma_t \bar{g}_t) = \tilde{x}_t - \gamma_t \bar{g}_t.$$

Hence the result. $\qquad\square$

### B.3 Non-convex convergence analysis

In this section, we provide the non-convex convergence analyses. Lemma 13 provides a one-step descent recurrence which leads to Theorem 1 and a key result for proving convergence. Based on this, in §B.3.1, §B.3.2, §B.3.3 we discuss the convergence of absolute compressors, $\delta$-contraction operators, and uncompressed SGD, respectively. In §B.3.4 we provide the convergence result for absolute compressors and $\delta$-contraction operators for an appropriate choice of step-size. The following lemma bounds the quantity $\mathbb{E}_t\|\frac{1}{n}\sum_{i=1}^{n}g_{i,t}\|^2$.

**Lemma 12.** *Let $f$ follow Assumption 4 and the stochastic noise, $\xi_{i,t}$ follow Assumption 3. Then we have*

$$\mathbb{E}_t\|\frac{1}{n}\sum_{i=1}^{n}g_{i,t}\|^2 \leq (1 + \frac{M(C+1)}{n})\|\nabla f(x_t)\|^2 + \frac{M\zeta^2 + \sigma^2}{n}. \tag{15}$$

*Proof.* Let the stochastic gradient, $g_{i,t}$ computed at $i^{th}$ worker at iteration $t$ follows $g_{i,t} = \nabla f_i(x_t) + \xi_{i,t}$ with $\mathbb{E}[\xi_{i,t}|x_t] = \mathbf{0}$. Hence, we have

$$\mathbb{E}_t \|\frac{1}{n}\sum_{i=1}^n g_{i,t}\|^2 \quad = \quad \mathbb{E}_t\|\frac{1}{n}\sum_{i=1}^n (\nabla f_i(x_t) + \xi_{i,t})\|^2$$

$$\overset{\mathbb{E}[\xi_{i,t}|x_t]=0}{=} \quad \|\nabla f(x_t)\|^2 + \mathbb{E}_t\|\frac{1}{n}\sum_{i=1}^n \xi_{i,t}\|^2$$

$$\overset{\mathbb{E}[\xi_{i,t}|x_t]=0}{=} \quad \|\nabla f(x_t)\|^2 + \frac{1}{n^2}\sum_{i=1}^n \mathbb{E}_t\|\xi_{i,t}\|^2$$

$$\overset{\text{By Assumption 3}}{\leq} \quad \|\nabla f(x_t)\|^2 + \frac{1}{n^2}\sum_{i=1}^n (M\|\nabla f_i(x_t)\|^2 + \sigma^2)$$

$$= \quad \|\nabla f(x_t)\|^2 + \frac{M}{n^2}\sum_{i=1}^n \|\nabla f_i(x_t) - \nabla f(x_t)\|^2 + \frac{M\|\nabla f(x_t)\|^2}{n} + \frac{\sigma^2}{n}$$

$$\overset{\text{By Assumption 4}}{\leq} \quad (1 + \frac{M}{n})\|\nabla f(x_t)\|^2 + \frac{M}{n}(C\|\nabla f(x_t)\|^2 + \zeta^2) + \frac{\sigma^2}{n}.$$

By rearranging the terms we get the result. $\qquad\square$

The following non-convex descent lemma is the key result used to establish convergence of both absolute compressors and $\delta$-contraction operators.

**Lemma 13.** *(**Non-convex descent lemma**) Let Assumptions 1, 3, and 4 hold. If $\{x_t\}_{t\geq 0}$ denote the iterates of Algorithm 1 for a constant step-size, $\gamma \leq \dfrac{n}{2L(M(C+1)+n)}$, then*

$$\mathbb{E}[f(\tilde{x}_{t+1})]] \leq \mathbb{E}[f(\tilde{x}_t)] - \frac{\gamma}{4}\mathbb{E}\|\nabla f(x_t)\|^2 + \frac{\gamma^2 L(M\zeta^2 + \sigma^2)}{2n} + \frac{\gamma L^2}{2n}\sum_{i=1}^n \mathbb{E}\|e_{i,t}\|^2. \tag{16}$$

*Proof.* By using the $L$-smoothness of $f$ and taking expectation we have

$$\mathbb{E}_t[f(\tilde{x}_{t+1})] \quad \leq \quad f(\tilde{x}_t) - \langle \nabla f(\tilde{x}_t), \mathbb{E}_t[\tilde{x}_{t+1} - \tilde{x}_t]\rangle + \frac{L}{2}\mathbb{E}_t\|\tilde{x}_{t+1} - \tilde{x}_t\|^2$$

$$= \quad f(\tilde{x}_t) - \gamma\langle \nabla f(\tilde{x}_t), \nabla f(x_t)\rangle + \frac{\gamma^2 L}{2}\mathbb{E}_t\|\frac{1}{n}\sum_{i=1}^n g_{i,t}\|^2$$

$$\overset{(15)}{\leq} \quad f(\tilde{x}_t) - \gamma\langle \nabla f(\tilde{x}_t), \nabla f(x_t)\rangle$$
$$+ \frac{\gamma^2 L}{2}\left((1 + \frac{M(C+1)}{n})\|\nabla f(x_t)\|^2 + \frac{M\zeta^2}{n} + \frac{\sigma^2}{n}\right)$$

$$\leq \quad f(\tilde{x}_t) - \gamma\|\nabla f(x_t)\|^2 + \gamma\langle \nabla f(x_t) - \nabla f(\tilde{x}_t), \nabla f(x_t)\rangle$$
$$+ \frac{\gamma^2 L(M(C+1)+n)}{2n}\|\nabla f(x_t)\|^2 + \frac{\gamma^2 L(M\zeta^2 + \sigma^2)}{2n}$$

$$\overset{(11)}{\leq} \quad f(\tilde{x}_t) - (\gamma - \frac{\gamma}{2} - \frac{\gamma^2 L(M(C+1)+n)}{2n})\|\nabla f(x_t)\|^2 +$$
$$\frac{\gamma\|\nabla f(x_t) - \nabla f(\tilde{x}_t)\|^2}{2} + \frac{\gamma^2 L(M\zeta^2 + \sigma^2)}{2n}$$

$$\overset{\substack{\text{By } L-\text{smoothness} \\ \text{and } \gamma \leq \frac{n}{2L(M(C+1)+n)}}}{\leq} \quad f(\tilde{x}_t) - \frac{\gamma\|\nabla f(x_t)\|^2}{4} + \frac{\gamma L^2\|x_t - \tilde{x}_t\|^2}{2} + \frac{\gamma^2 L(M\zeta^2 + \sigma^2)}{2n}$$

$$= \quad f(\tilde{x}_t) - \frac{\gamma\|\nabla f(x_t)\|^2}{4} + \frac{\gamma L^2\|\bar{e}_t\|^2}{2} + \frac{\gamma^2 L(M\zeta^2 + \sigma^2)}{2n}$$

$$\overset{(12)}{\leq} \quad f(\tilde{x}_t) - \frac{\gamma\|\nabla f(x_t)\|^2}{4} + \frac{\gamma L^2\frac{1}{n}\sum_{i=1}^n\|e_{i,t}\|^2}{2} + \frac{\gamma^2 L(M\zeta^2 + \sigma^2)}{2n}.$$

Taking total expectation yields the lemma. □

*Remark* 10. Rearranging the terms in Lemma 13, performing telescopic sum, and noting that $\zeta = 0$ for $n = 1$, we get the result in Theorem 1.

### B.3.1 Absolute compressors

**Theorem.** *6 (Non-convex convergence of absolute compressors) Let Assumptions 1, 2, 3, and 4 hold. Then the iterates, $\{x_t\}_{t\geq 0}$ of Algorithm 1 with an absolute compressor, $\mathcal{C}$ and a constant step-size, $\gamma \leq \frac{n}{2L(M(C+1)+n)}$, follow*

$$\frac{1}{T} \sum_{t=0}^{T-1} \mathbb{E}\|\nabla f(x_t)\|^2 \leq \frac{4(f(x_0) - f^\star)}{\gamma T} + \frac{2\gamma L(M\zeta^2 + \sigma^2)}{n} + 2\gamma^2 L^2 v^2.$$

*Proof.* By using Lemma 13, we have

$$
\begin{aligned}
\mathbb{E}[f(\tilde{x}_{t+1})] \quad &\leq \quad \mathbb{E}[f(\tilde{x}_t)] - \frac{\gamma \mathbb{E}\|\nabla f(x_t)\|^2}{4} + \frac{\gamma L^2 \frac{1}{n}\sum_{i=1}^{n} \mathbb{E}\|e_{i,t}\|^2}{2} + \frac{\gamma^2 L(M\zeta^2 + \sigma^2)}{2n} \\
&\overset{\text{Remark 5}}{\leq} \mathbb{E}[f(\tilde{x}_t)] - \frac{\gamma \mathbb{E}\|\nabla f(x_t)\|^2}{4} + \frac{\gamma^3 L^2 v^2}{2} + \frac{\gamma^2 L(M\zeta^2 + \sigma^2)}{2n}.
\end{aligned}
$$

By taking summation over the iterates, we get

$$
\begin{aligned}
\frac{1}{T} \sum_{t=0}^{T-1} \mathbb{E}\|\nabla f(x_t)\|^2 &\leq \frac{4\sum_{t=0}^{T-1}(\mathbb{E}[f(\tilde{x}_t)] - \mathbb{E}[f(\tilde{x}_{t+1})])}{\gamma T} + \frac{2\gamma L(M\zeta^2 + \sigma^2)}{n} + 2\gamma^2 L^2 v^2 \\
&\leq \frac{4(f(x_0) - f^\star)}{\gamma T} + \frac{2\gamma L(M\zeta^2 + \sigma^2)}{n} + 2\gamma^2 L^2 v^2.
\end{aligned}
$$

Hence the result. □

### B.3.2 $\delta$-contraction operators

We now provide an error-bound for $\delta$-contraction operators, which is an extension of the single node case in [54].

**Lemma 14.** *Let $f$ follow Assumption 4 and the stochastic noise follow Assumptions 3. Define $e_{i,t}$ as in Algorithm 1. Then by using a $\delta$-compressor, $\mathcal{C}$, with a constant step-size, $\gamma \leq \frac{1}{2L(2/\delta+M)\sqrt{C+1}}$, we have*

$$\sum_{t=0}^{T} \left[ \frac{1}{n} \sum_{i=1}^{n} \mathbb{E}\|e_{i,t}\|^2 \right] \leq \frac{1}{4L^2} \sum_{t=0}^{T} \mathbb{E}\|\nabla f(x_t)\|^2 + \frac{2\gamma^2(T+1)}{\delta}\left(\left(\frac{2}{\delta} + M\right)\zeta^2 + \sigma^2\right). \quad (17)$$

*Proof.* We note that the compression operator, $\mathcal{C}$ and the stochastic noise, $\xi_{i,t}$ are independent of each other. Therefore, by taking expectation on the randomness of the compression operator, $\mathcal{C}$ in the following expression we have

$$
\begin{aligned}
\frac{1}{n} \sum_{i=1}^{n} \mathbb{E}_{\mathcal{C}}\|e_{i,t+1}\|^2 \quad &= \quad \frac{1}{n} \sum_{i=1}^{n} \mathbb{E}_{\mathcal{C}}\|e_{i,t} + \gamma g_{i,t} - \gamma\mathcal{C}(\frac{e_{i,t}}{\gamma} + g_{i,t})\|^2 \\
&\overset{\text{By (7)}}{\leq} \frac{1}{n} \sum_{i=1}^{n} \gamma^2(1 - \delta)\|\frac{e_{i,t}}{\gamma} + g_{i,t}\|^2,
\end{aligned}
$$

which further by taking expectation conditioned on $x_t$ becomes

$$\frac{1}{n}\sum_{i=1}^{n}\mathbb{E}\left(\mathbb{E}_{\mathcal{C}}\|e_{i,t+1}\|^2|x_t\right) \overset{\mathbb{E}[\xi_{i,t}|x_t]=0}{\leq} \frac{(1-\delta)}{n}\sum_{i=1}^{n}\|e_{i,t}+\gamma\nabla f_i(x_t)\|^2 + \frac{(1-\delta)}{n}\sum_{i=1}^{n}\gamma^2\mathbb{E}\left[\|\xi_{i,t}\|^2|x_t\right]$$

$$\overset{\text{Assumption 3}}{\leq} \frac{(1-\delta)}{n}\sum_{i=1}^{n}\|e_{i,t}+\gamma\nabla f_i(x_t)\|^2 + \frac{(1-\delta)\gamma^2}{n}\sum_{i=1}^{n}\left(M\|\nabla f_i(x_t)\|^2+\sigma^2\right)$$

$$\overset{(10)}{\leq} \frac{(1-\delta)(1+\rho)}{n}\sum_{i=1}^{n}\|e_{i,t}\|^2 + \frac{(1-\delta)(1+\rho^{-1}+M)\gamma^2}{n}\sum_{i=1}^{n}\|\nabla f_i(x_t)\|^2$$

$$+(1-\delta)\gamma^2\sigma^2$$

$$\overset{\text{Assumption 4}}{\leq} \frac{(1-\delta)(1+\rho)}{n}\sum_{i=1}^{n}\|e_{i,t}\|^2 + \left((1-\delta)(1+\rho^{-1}+M)\gamma^2(C+1)\right)\|\nabla f(x_t)\|^2$$

$$+\left((1-\delta)(1+\rho^{-1}+M)\gamma^2\zeta^2\right)+(1-\delta)\gamma^2\sigma^2$$

$$\leq \frac{(1-\delta)(1+\rho)}{n}\sum_{i=1}^{n}\|e_{i,t}\|^2$$

$$+\gamma^2\left((1+\rho^{-1}+M)(C+1)\|\nabla f(x_t)\|^2+(1+\rho^{-1}+M)\zeta^2+\sigma^2\right).$$

By unrolling the recurrence, taking total expectation, setting $\rho = \frac{\delta}{2(1-\delta)}$, such that $(1+\rho^{-1}) = \frac{2-\delta}{\delta} \leq \frac{2}{\delta}$ and $(1-\delta)(1+\rho) \leq (1-\frac{\delta}{2})$, and using the fact that $e_{i,0}=0$, for all $i$, we find

$$\frac{1}{n}\sum_{i=1}^{n}\mathbb{E}\|e_{i,t+1}\|^2 \leq \gamma^2\sum_{i=0}^{t}[(1-\delta)(1+\rho)]^{t-i}\left(\left(\frac{2}{\delta}+M\right)(C+1)\mathbb{E}\|\nabla f(x_i)\|^2+\left(\frac{2}{\delta}+M\right)\zeta^2+\sigma^2\right)$$

$$\leq \gamma^2\sum_{i=0}^{t}(1-\frac{\delta}{2})^{t-i}\left(\left(\frac{2}{\delta}+M\right)(C+1)\mathbb{E}\|\nabla f(x_i)\|^2+\left(\frac{2}{\delta}+M\right)\zeta^2+\sigma^2\right).$$

Finally,

$$\sum_{t=0}^{T}\left[\frac{1}{n}\sum_{i=1}^{n}\mathbb{E}\|e_{i,t}\|^2\right] = \gamma^2\sum_{t=0}^{T}\sum_{i=0}^{t-1}(1-\frac{\delta}{2})^{t-1-i}\left(\left(\frac{2}{\delta}+M\right)(C+1)\mathbb{E}\|\nabla f(x_i)\|^2+\left(\frac{2}{\delta}+M\right)\zeta^2+\sigma^2\right)$$

$$\leq \gamma^2\sum_{t=0}^{T-1}\sum_{j=0}^{T-t-1}(1-\frac{\delta}{2})^{j}\left(\left(\frac{2}{\delta}+M\right)(C+1)\mathbb{E}\|\nabla f(x_t)\|^2+\left(\frac{2}{\delta}+M\right)\zeta^2+\sigma^2\right)$$

$$\leq \gamma^2\sum_{t=0}^{T-1}\left(\left(\frac{2}{\delta}+M\right)(C+1)\mathbb{E}\|\nabla f(x_t)\|^2+\left(\frac{2}{\delta}+M\right)\zeta^2+\sigma^2\right)\sum_{j=0}^{\infty}(1-\frac{\delta}{2})^{j}$$

$$= \gamma^2\sum_{t=0}^{T-1}\left(\frac{2}{\delta}\right)\left(\left(\frac{2}{\delta}+M\right)(C+1)\mathbb{E}\|\nabla f(x_t)\|^2+\left(\frac{2}{\delta}+M\right)\zeta^2+\sigma^2\right)$$

$$\leq \gamma^2\sum_{t=0}^{T}\left(\frac{2}{\delta}\right)\left(\left(\frac{2}{\delta}+M\right)(C+1)\mathbb{E}\|\nabla f(x_t)\|^2+\left(\frac{2}{\delta}+M\right)\zeta^2+\sigma^2\right)$$

$$= \sum_{t=0}^{T}\left(\gamma^2\left(\frac{2}{\delta}\right)\left(\frac{2}{\delta}+M\right)(C+1)\mathbb{E}\|\nabla f(x_t)\|^2\right)+\sum_{t=0}^{T}\frac{2\gamma^2}{\delta}\left(\left(\frac{2}{\delta}+M\right)\zeta^2+\sigma^2\right).$$

Choosing $\gamma \leq \frac{1}{2L(2/\delta+M)\sqrt{C+1}}$, we get $\gamma^2\left(\frac{2}{\delta}\right)\left(\frac{2}{\delta}+M\right) \leq \frac{1}{4L^2(C+1)}$. Combining all together we have

$$\sum_{t=0}^{T}\left[\frac{1}{n}\sum_{i=1}^{n}\mathbb{E}\|e_{i,t}\|^2\right] \leq \frac{1}{4L^2}\sum_{t=0}^{T}\mathbb{E}\|\nabla f(x_t)\|^2 + \frac{2\gamma^2(T+1)}{\delta}\left(\left(\frac{2}{\delta}+M\right)\zeta^2+\sigma^2\right).$$

Hence the result. □

By using the previous bound, we now provide the non-convex convergence result for $\delta$-contraction operators.

**Theorem.** *7 (Non-convex convergence of $\delta$-contraction operators) Let Assumptions 1, 2, 3, and 4 hold. Then the iterates, $\{x_t\}_{t\geq 0}$ of Algorithm 1 with a $\delta$-contraction operator and a constant step-size $\gamma \leq \min\{\frac{n}{2L(M(C+1)+n)}, \frac{1}{2L(2/\delta+M)\sqrt{C+1}}\}$ follow*

$$\frac{1}{T}\sum_{t=0}^{T-1}\mathbb{E}\|\nabla f(x_t)\|^2 \leq \frac{8(f(x_0)-f^\star)}{\gamma T} + \frac{4\gamma L(M\zeta^2+\sigma^2)}{n} + \frac{8\gamma^2 L^2}{\delta}\left(\left(\frac{2}{\delta}+M\right)\zeta^2+\sigma^2\right).$$

*Proof.* Summing over the iterates $t=0$ to $t=T-1$ in (16) of Lemma 13, we have

$$\mathbb{E}[f(\tilde{x}_T)] \leq f(x_0) - \frac{\sum_{t=0}^{T-1}\gamma\mathbb{E}\|\nabla f(x_t)\|^2}{4} + \frac{\gamma L^2 \sum_{t=0}^{T-1}\frac{1}{n}\sum_{i=1}^{n}\mathbb{E}\|e_{i,t}\|^2}{2} + \sum_{t=0}^{T-1}\frac{\gamma^2 L(M\zeta^2+\sigma^2)}{2n}$$

$$\overset{(17)}{\leq} f(x_0) - (\frac{\gamma}{4}-\frac{\gamma}{8})\sum_{t=0}^{T-1}\mathbb{E}\|\nabla f(x_t)\|^2 + \frac{\gamma^3 L^2 T}{\delta}\left(\left(\frac{2}{\delta}+M\right)\zeta^2+\sigma^2\right) + \frac{\gamma^2 TL(M\zeta^2+\sigma^2)}{2n}.$$

Rearranging, we get

$$\frac{1}{T}\sum_{t=0}^{T-1}\mathbb{E}\|\nabla f(x_t)\|^2 \leq \frac{8(f(x_0)-\mathbb{E}[f(\tilde{x}_t)])}{\gamma T} + \frac{4\gamma L(M\zeta^2+\sigma^2)}{n} + \frac{8\gamma^2 L^2}{\delta}\left(\left(\frac{2}{\delta}+M\right)\zeta^2+\sigma^2\right)$$

$$\leq \frac{8(f(x_0)-f^\star)}{\gamma T} + \frac{4\gamma L(M\zeta^2+\sigma^2)}{n} + \frac{8\gamma^2 L^2}{\delta}\left(\left(\frac{2}{\delta}+M\right)\zeta^2+\sigma^2\right).$$

Hence the result. $\square$

### B.3.3 Uncompressed SGD

We provide the convergence result of no-compression SGD (Algorithm 1 with an identity compressor, i.e., $\mathcal{C}(x) = x$ for all $x \in \mathbb{R}^d$).

**Theorem 8.** *(Non-convex convergence of SGD) Let Assumptions 1, 2, 3, and 4 hold. Then the iterates, $\{x_t\}_{t\geq 0}$ of Algorithm 1 by using an identity compressor ($\mathcal{C}(x) = x$, for all $x \in \mathbb{R}^d$) with a constant step-size, $\gamma \leq \frac{n}{L(M(C+1)+n)}$ follow*

$$\frac{1}{T}\sum_{t=0}^{T-1}\mathbb{E}\|\nabla f(x_t)\|^2 \leq \frac{2(f(x_0)-f^\star)}{\gamma T} + \frac{\gamma L(M\zeta^2+\sigma^2)}{n}.$$

*Proof.* We use the $L$-smoothness of $f$ to find

$$\mathbb{E}_t[f(x_{t+1})] \quad \leq \quad f(x_t) - \langle\nabla f(x_t), \mathbb{E}_t[x_{t+1}-x_t]\rangle + \frac{L}{2}\mathbb{E}_t\|x_{t+1}-x_t\|^2$$

$$= \quad f(x_t) - \gamma\langle\nabla f(x_t), \mathbb{E}_t[\bar{g}_t]\rangle + \frac{\gamma^2 L}{2}\mathbb{E}_t\|\bar{g}_t\|^2$$

$$= \quad f(x_t) - \gamma\|\nabla f(x_t)\|^2 + \frac{\gamma^2 L}{2}\mathbb{E}_t\|\frac{1}{n}\sum_{i=1}^{n}g_{i,t}\|^2$$

$$\overset{(15)}{\leq} \quad f(x_t) - \gamma\|\nabla f(x_t)\|^2 + \frac{\gamma^2 L}{2}\left((1+\frac{M(C+1)}{n})\|\nabla f(x_t)\|^2 + \frac{M\zeta^2}{n} + \frac{\sigma^2}{n}\right)$$

$$= \quad f(x_t) - \gamma\left(1-\frac{\gamma L(M(C+1)+n)}{2n}\right)\|\nabla f(x_t)\|^2 + \frac{\gamma^2 L(M\zeta^2+\sigma^2)}{2n}$$

$$\overset{\gamma\leq\frac{n}{L(M(C+1)+n)}}{\leq} \quad f(x_t) - \frac{\gamma}{2}\|\nabla f(x_t)\|^2 + \frac{\gamma^2 L(M\zeta^2+\sigma^2)}{2n}.$$

By summing over the iterates and taking total expectation, we get

$$\frac{1}{T}\sum_{t=0}^{T-1}\mathbb{E}\|\nabla f(x_t)\|^2 \leq \frac{2(f(x_0)-f^\star)}{\gamma T} + \frac{\gamma L(M\zeta^2+\sigma^2)}{n}.$$

Hence the result. $\square$

### B.3.4 Final convergence result

From Remark 9, the following corollary describes the $\mathcal{O}(1/\sqrt{nT})$ convergence with an appropriate step-size for absolute compressors and $\delta$-contraction operators.

**Corollary 1.** *Let Assumptions 1, 2, 3, and 4 hold with $M\zeta^2 + \sigma^2 > 0$ and let $\{x_t\}_{t\geq 0}$ denote the iterates of algorithm 1. Then, if*
$\bullet$ $\mathcal{C}$ *is an absolute compressor, we have*

$$\frac{1}{T}\sum_{t=0}^{T-1}\mathbb{E}\|\nabla f(x_t)\|^2 = \mathcal{O}\left(\frac{\sqrt{L(M\zeta^2+\sigma^2)(f(x_0)-f^\star)}}{\sqrt{nT}} + \frac{L((\frac{M}{n}(C+1)+1)+\frac{nv^2}{M\zeta^2+\sigma^2})(f(x_0)-f^\star)}{T}\right).$$

$\bullet$ $\mathcal{C}$ *is a $\delta$-contraction operator, we have*

$$\frac{1}{T}\sum_{t=0}^{T-1}\mathbb{E}\|\nabla f(x_t)\|^2 = \mathcal{O}\left(\frac{\sqrt{(L(M\zeta^2+\sigma^2))(f(x_0)-f^\star)}}{\sqrt{nT}} + \frac{L\left(\max\{\frac{M}{n}(C+1)+1\},(\frac{1}{\delta}+M)\sqrt{C+1}\}+\frac{n\left((1+M\delta)\zeta^2+\delta\sigma^2\right)}{\delta^2(M\zeta^2+\sigma^2)}\right)(f(x_0)-f^\star)}{T}\right).$$

$\bullet$ $\mathcal{C}$ *is the identity compressor, we have*

$$\frac{1}{T}\sum_{t=0}^{T-1}\mathbb{E}\|\nabla f(x_t)\|^2 = \mathcal{O}\left(\frac{\sqrt{L(M\zeta^2+\sigma^2)(f(x_0)-f^\star)}}{\sqrt{nT}} + \frac{L(\frac{M}{n}(C+1)+1)(f(x_0)-f^\star)}{T}\right).$$

*Proof.* Invoking Lemma 7 in Theorem 6 and Theorem 7, and Lemma 6 in Theorem 8 we get the results. $\qquad\square$

We note that the above results are for the cases with $M\zeta^2 + \sigma^2 > 0$. If $M\zeta^2 + \sigma^2 = 0$, i.e. a non-stochastic setting, then one can derive the convergence result using Lemma 8.

While compression does not affect the slower decaying $\mathcal{O}(1/\sqrt{nT})$ term for both absolute compressors and $\delta$-contraction operators, we observe $\delta$-contraction operators have $1/\delta^2$ dependence in the $\mathcal{O}(1/T)$ term when $\zeta \neq 0$ (heterogeneous data). Therefore, in this setting, the Top-$k$ sparsifier has $d^2/k^2$ in the numerator of $\mathcal{O}(1/T)$ term. On the other hard, hard-threshold has $d\lambda^2$ in the numerator of $\mathcal{O}(1/T)$ term even when $\zeta \neq 0$, and thus has a significantly better dependence on $d$.

### B.4 Convex convergence analysis

In this Section, we provide convergence results for *distributed compressed SGD* with *absolute compressors* and an *EF* where the loss function on each worker $f_i$ is $\mu$-strongly convex with $\mu \geq 0$ (see Assumption 5). Our analysis is inspired by the proof techniques in [54] which analyzes an EF SGD with $\delta$-contraction operators in the single node ($n = 1$) case. [10] extended this analysis to the distributed ($n > 1$) case for $\delta$-contraction operators.

We start with the following key result by Nesterov [41] for convex and smooth functions.

**Lemma 15.** *Let $f_i$ follow Assumptions 1 and Assumption 5 with $\mu \geq 0$, then*

$$\|\nabla f_i(y) - \nabla f_i(x)\|^2 \leq 2L(f_i(y) - f_i(x) - \langle\nabla f_i(x), y - x\rangle), \qquad \forall x, y \in \mathbb{R}^d. \tag{18}$$

We start with the convex decent lemma from [10]. For completeness, we also provide the proof.

**Lemma 16.** *(Convex descent lemma) (Lemma 21 in [10]) Let Assumptions 1, 2, 3, and 5 hold. Denote $D := \frac{1}{n}\sum_{i=1}^{n}\|\nabla f_i(x^\star)\|^2$. If $\gamma_t \leq \frac{1}{4L(1+2M/n)}$, for all $t \geq 0$, then the iterates, $\{\tilde{x}_t\}_{t\geq 0}$ of Algorithm 1 follow*

$$\mathbb{E}_t\|\tilde{x}_{t+1} - x^\star\|^2 \leq (1 - \frac{\mu\gamma_t}{2})\|\tilde{x}_t - x^\star\|^2 - \frac{\gamma_t}{2}[f(x_t) - f^\star] + 3L\gamma_t\|x_t - \tilde{x}_t\|^2 + (\gamma_t^2)\frac{\sigma^2 + 2MD}{n}.$$

*Proof.* We have

$$\|\tilde{x}_{t+1} - x^\star\|^2 \overset{\text{Lemma 11}}{=} \|\tilde{x}_t - x^\star\|^2 - 2\gamma_t\langle\bar{g}_t, \tilde{x}_t - x^\star\rangle + \gamma_t^2\|\bar{g}_t\|^2$$
$$= \|\tilde{x}_t - x^\star\|^2 - 2\gamma_t\langle\bar{g}_t, x_t - x^\star\rangle + \gamma_t^2\|\bar{g}_t\|^2 + 2\gamma_t\langle\bar{g}_t, x_t - \tilde{x}_t\rangle.$$

Therefore,

$$\mathbb{E}_t \|\tilde{x}_{t+1} - x^\star\|^2 = \|\tilde{x}_t - x^\star\|^2 - 2\gamma_t \langle \mathbb{E}_t[\bar{g}_t], x_t - x^\star \rangle + \gamma_t^2 \mathbb{E}_t \|\bar{g}_t\|^2 + 2\gamma_t \langle \mathbb{E}_t[\bar{g}_t], x_t - \tilde{x}_t \rangle$$

$$= \|\tilde{x}_t - x^\star\|^2 - 2\gamma_t \langle \nabla f(x_t), x_t - x^\star \rangle + \gamma_t^2 \mathbb{E}_t \|\bar{g}_t\|^2 + 2\gamma_t \langle \nabla f(x_t), x_t - \tilde{x}_t \rangle . \tag{19}$$

First, we bound $2 \langle \nabla f(x_t), x_t - \tilde{x}_t \rangle$. We use Young's inequality (11) with $\rho = \frac{1}{2L}$ and get

$$2 \langle \nabla f(x_t), x_t - \tilde{x}_t \rangle \leq \frac{1}{2L} \|\nabla f(x_t)\|^2 + 2L \|x_t - \tilde{x}_t\|^2$$

$$\overset{(18), \nabla f(x^\star)=0}{\leq} f(x_t) - f(x^\star) + 2L \|x_t - \tilde{x}_t\|^2 . \tag{20}$$

Next, we bound $-2 \langle \nabla f(x_t), x_t - x^\star \rangle$. We use the $\mu$-strong convexity of $f$ to find

$$-2 \langle \nabla f(x_t), x_t - x^\star \rangle \leq 2(f(x^\star) - f(x_t)) - \mu \|x_t - x^\star\|^2 . \tag{21}$$

However, since we want to work with $\|\tilde{x}_t - x^\star\|^2$ instead of $\|x_t - x^\star\|^2$, we get rid of $\|x_t - x^\star\|^2$ using (10) with $\rho = 1$ as

$$\|x_t - x^\star\|^2 \geq \frac{1}{2} \|\tilde{x}_t - x^\star\|^2 - \|x_t - \tilde{x}_t\|^2 .$$

Substituting this in Equation (21), we get

$$-2 \langle \nabla f(x_t), x_t - x^\star \rangle \leq 2(f(x^\star) - f(x_t)) - \frac{\mu}{2} \|\tilde{x}_t - x^\star\|^2 + \mu \|x_t - \tilde{x}_t\|^2 . \tag{22}$$

Finally, we bound $\mathbb{E}_t \|\bar{g}_t\|^2$ as

$$\mathbb{E}_t \| \frac{1}{n} \sum_{i=1}^{n} g_{i,t} \|^2 = \mathbb{E} \left[ \| \frac{1}{n} \sum_{i=1}^{n} (\nabla f_i(x_t) + \xi_{i,t}) \|^2 | x_t \right]$$

$$= \mathbb{E} \left[ \| \nabla f(x_t) + \frac{1}{n} \sum_{i=1}^{n} \xi_{i,t} \|^2 | x_t \right]$$

$$\overset{\mathbb{E}[\xi_{i,t}|x_t]=0}{=} \|\nabla f(x_t)\|^2 + \mathbb{E} \left[ \| \frac{1}{n} \sum_{i=1}^{n} \xi_{i,t} \|^2 | x_t \right]$$

$$\overset{\mathbb{E}[\xi_{i,t}|x_t]=0}{=} \|\nabla f(x_t)\|^2 + \frac{1}{n^2} \sum_{i=1}^{n} \mathbb{E} \left[ \|\xi_{i,t}\|^2 | x_t \right]$$

$$\overset{\text{Assumption 3}}{\leq} \|\nabla f(x_t)\|^2 + \frac{1}{n^2} \sum_{i=1}^{n} (M\|\nabla f_i(x_t)\|^2 + \sigma^2)$$

$$= \|\nabla f(x_t)\|^2 + \frac{M}{n^2} \sum_{i=1}^{n} \|\nabla f_i(x_t) - \nabla f_i(x^\star) + \nabla f_i(x^\star)\|^2 + \frac{\sigma^2}{n}$$

$$\leq \|\nabla f(x_t)\|^2 + \frac{2M}{n^2} \sum_{i=1}^{n} \left( \|\nabla f_i(x_t) - \nabla f_i(x^\star)\|^2 + \|\nabla f_i(x^\star)\|^2 \right)$$

$$+ \frac{\sigma^2}{n}$$

$$\overset{(18), D=\frac{1}{n}\sum_{i=1}^{n}\|\nabla f_i(x^\star)\|^2}{\leq} \|\nabla f(x_t)\|^2 + \frac{2M}{n^2} \sum_{i=1}^{n} 2L[f_i(x_t) - f_i(x^\star) - \langle \nabla f_i(x^\star), x_t - x^\star \rangle]$$

$$+ \frac{2MD}{n} + \frac{\sigma^2}{n} \tag{23}$$

$$\overset{\nabla f(x^\star)=0}{=} \|\nabla f(x_t) - \nabla f(x^\star)\|^2 + \frac{4LM}{n} (f(x_t) - f(x^\star)) + \frac{2MD + \sigma^2}{n}$$

$$\overset{(18), \nabla f(x^\star)=0}{\leq} 2L \left( 1 + \frac{2M}{n} \right) (f(x_t) - f(x^\star)) + \frac{2MD + \sigma^2}{n} . \tag{24}$$

We now substitute (20), (22), and (24) in (19) to get

$$\mathbb{E}_t\|\tilde{x}_{t+1} - x^\star\|^2 = \|\tilde{x}_t - x^\star\|^2 - 2\gamma_t \langle \nabla f(x_t), x_t - x^\star \rangle + \gamma_t^2 \mathbb{E}_t\|\bar{g}_t\|^2 + 2\gamma_t \langle \nabla f(x_t), x_t - \tilde{x}_t \rangle$$

$$\leq \left(1 - \frac{\mu\gamma_t}{2}\right)\|\tilde{x}_t - x^\star\|^2 - \gamma_t\left(1 - \gamma_t \cdot 2L\left(1 + \frac{2M}{n}\right)\right)(f(x_t) - f(x^\star))$$

$$+ \gamma_t(2L + \mu)\|x_t - \tilde{x}_t\|^2 + \gamma_t^2\frac{2MD + \sigma^2}{n}.$$

Choosing $\gamma_t \leq \frac{1}{4L(1+2M/n)}$ gives the desired result. $\qquad\square$

Next, we give the convex convergence result of *distributed EF SGD* with *absolute compressors*.

### B.4.1  Absolute compressors

The next theorem combines the results of Theorems 4 and 5 from the main paper. We present them as a single theorem (Theorem 9) to keep the structure of the proofs simple.

**Theorem 9.** *Let Assumptions 1, 2, 3, and 5 hold. Denote $D := \frac{1}{n}\sum_{i=1}^{n}\|\nabla f_i(x^\star)\|^2$, and $R_0 = \|x_0 - x^\star\|^2$. Then the iterates, $\{x_t\}_{t\geq 0}$ of Algorithm 1 with an absolute compressor, $\mathcal{C}_v$ have the following convergence rates if Assumption 5 is satisfied with the following choices of the parameters:*

*i) (**Theorem 4**) If $\mu > 0$, a constant step-size $\{\gamma_t = \gamma\}_{t\geq 0}$, with $\gamma \leq \frac{1}{4L(1+2M/n)}$ is chosen as in Lemma 10 and weights $\{w_t = (1 - \mu\gamma/2)^{-(t+1)}\}_{t\geq 0}$ then*

$$\mathbb{E}[f(\bar{x}_T)] - f^\star = \tilde{\mathcal{O}}\left(L(1 + M/n)R_0\exp\left[-\frac{\mu T}{8L(1+2M/n)}\right] + \frac{\sigma^2 + MD}{\mu n T} + \frac{Lv^2}{\mu^2 T^2}\right).$$

*ii) (**Theorem 5**) If $\mu = 0$, a constant step-size $\{\gamma_t = \gamma\}_{t\geq 0}$, with $\gamma \leq \frac{1}{4L(1+2M/n)}$ is chosen as in Lemma 7 and weights $\{w_t = 1\}_{t\geq 0}$ then*

$$\mathbb{E}[f(\bar{x}_T)] - f^\star = \mathcal{O}\left(\frac{\sqrt{(\sigma^2 + MD)R_0}}{\sqrt{nT}} + \frac{\left(\frac{nLv^2}{\sigma^2 + MD} + L(1 + M/n)\right)R_0}{T}\right).$$

*iii) If $\mu > 0$, step-sizes $\{\gamma_t = \frac{4}{\mu(\phi+t)}\}_{t\geq 0}$, and weights $\{w_t = \phi + t\}_{t\geq 0}$, respectively with $\phi = \frac{16L}{\mu}(1 + \frac{2M}{n})$ then*

$$\mathbb{E}[f(\bar{x}_T)] - f^\star = \mathcal{O}\left(\frac{\sigma^2 + MD}{\mu n T} + \frac{\mu L^2(1 + M/n)^2 R_0 + Lv^2\ln(T)}{\mu^2 T^2}\right).$$

*In the above, $\bar{x}_T = \frac{1}{W_T}\sum_{t=0}^{T}w_t x_t$, and $W_T = \sum_{t=0}^{T}w_t$.*

*Proof.* By using Lemma 11 in Lemma 16, and taking total-expectation over all the previous iterates, we have

$$\mathbb{E}\|\tilde{x}_{t+1} - x^\star\|^2 \leq (1 - \frac{\mu\gamma_t}{2})\mathbb{E}\|\tilde{x}_t - x^\star\|^2 - \frac{\gamma_t}{2}\mathbb{E}[f(x_t) - f^\star] + 3L\gamma_t\mathbb{E}\|\bar{e}_t\|^2 + \gamma_t^2(\frac{\sigma^2 + 2MD}{n})$$

$$\overset{(12)}{\leq} (1 - \frac{\mu\gamma_t}{2})\mathbb{E}\|\tilde{x}_t - x^\star\|^2 - \frac{\gamma_t}{2}\mathbb{E}[f(x_t) - f^\star] + 3L\gamma_t\sum_{i=1}^{n}\frac{1}{n}\mathbb{E}\|e_{i,t}\|^2 \qquad (25)$$

$$+ \gamma_t^2(\frac{\sigma^2 + 2MD}{n}) \qquad (26)$$

$$\overset{\text{Remark 5}}{\leq} (1 - \frac{\mu\gamma_t}{2})\mathbb{E}\|\tilde{x}_t - x^\star\|^2 - \frac{\gamma_t}{2}\mathbb{E}[f(x_t) - f^\star] + 3L\gamma_t^3 v^2 + \gamma_t^2(\frac{\sigma^2 + 2MD}{n}).$$

Rearranging, we get

$$\mathbb{E}[f(x_t)] - f^\star \leq \frac{2}{\gamma_t}(1 - \frac{\mu\gamma_t}{2})\mathbb{E}\|\tilde{x}_t - x^\star\|^2 - \frac{2}{\gamma_t}\mathbb{E}\|\tilde{x}_{t+1} - x^\star\|^2 + \gamma_t\frac{2\sigma^2 + 4MD}{n} + 6L\gamma_t^2 v^2. \quad (27)$$

With $r_t = 2\mathbb{E}\|\tilde{x}_t - x^\star\|^2$, $a = \frac{\mu}{2}$, $c = \frac{2\sigma^2 + 4MD}{n}$, $b = 6Lv^2$, we can see the RHS as $\frac{1}{\gamma_t}(1 - a\gamma_t)r_t - \frac{1}{\gamma_t}r_{t+1} + c\gamma_t + b\gamma_t^2$. Thus, we use Lemma 10 and Lemma 9 to get the first and the third result respectively. Note that to get the LHS, we use the convexity of $f$ as $\frac{1}{W_T}\sum_{t=0}^{T} w_t f(x_t) \geq f(\bar{x}_T)$. Finally, to get the second result, we substitute $\mu = 0$ in Equation (27) and perform telescopic sum to get

$$\frac{\sum_{t=0}^{T}\mathbb{E}[f(x_t)]}{T+1} - f^\star \leq \frac{2\|x_0 - x^\star\|^2}{\gamma(T+1)} + \frac{2\sigma^2 + 4MD}{n}\gamma + 6Lv^2\gamma^2.$$

We now use Lemma 7 and convexity of $f$ to arrive at the desired result. Similarly, for the result of Remark 7, we use Lemma 8.

$\square$

### B.4.2 $\delta$-contraction operators

The rates for $\delta$-contraction operators is based on [22], except we consider a slightly different set of assumptions. Below we provide the sketch of the proof.

First, using equation (18), we can have

$$\begin{aligned}
\frac{1}{n}\sum_{i=1}^{n}\|\nabla f_i(x_t)\|^2 &= \frac{1}{n}\sum_{i=1}^{n}\|\nabla f_i(x_t)\|^2 \\
&= \frac{1}{n}\sum_{i=1}^{n}\|\nabla f_i(x_t) - \nabla f_i(x^\star) + \nabla f_i(x^\star)\|^2 \\
&\leq \frac{2}{n}\sum_{i=1}^{n}\|\nabla f_i(x_t) - \nabla f_i(x^\star)\|^2 + \frac{2}{n}\sum_{i=1}^{n}\|\nabla f_i(x^\star)\|^2 \\
&\overset{(18)}{\leq} 4L(f(x_t) - f^\star) + 2D.
\end{aligned} \tag{28}$$

Second, from Assumption 3, we have

$$\begin{aligned}
\frac{1}{n}\sum_{i=1}^{n}\mathbb{E}[\|\xi_{i,t}\|^2 \mid x_t] &\leq \frac{M}{n}\sum_{i=1}^{n}\|\nabla f_i(x_t)\|^2 + \sigma^2 \\
&\overset{(28)}{\leq} 4LM(f(x_t) - f^\star) + 2MD + \sigma^2.
\end{aligned} \tag{29}$$

Third, from (24), we have

$$\mathbb{E}\left[\|\frac{1}{n}\sum_{i=1}^{n} g_{i,t}\|^2 | x_t\right] \leq 2L\left(1 + \frac{2M}{n}\right)(f(x_t) - f(x^\star)) + \frac{2MD + \sigma^2}{n}. \tag{30}$$

Using (28), (29), and (30), we can show that Assumption 3.3 in [22] is satisfied with $A = 2L$, $D_1 = 2D$, $\tilde{A} = 2LM$, $\tilde{D}_1 = 2MD + \sigma^2$, $A' = L(1 + \frac{2M}{n})$, $D_1' = \frac{2MD+\sigma^2}{n}$, $\rho_1 = \rho_2 = 1$, and all the other quantities as zero. Then, using Lemma G.1 in [22] with $\gamma \leq \frac{\delta}{8L\sqrt{3(2+M\delta)}}$, we can show that Assumption 3.4 in [22] is satisfied with $F_1 = 0$, $F_2 = 0$, and $D_3 = \frac{6L\gamma}{\delta^2}\left(D(4 + 2M\delta) + \delta\sigma^2\right)$. We subsequently use (25), followed by Lemma 10 for the strongly-convex case (Remark 6), and Lemma 7 for the convex case (Remark 8).

## B.5 Comparison against unbiased compressors

Till now, we have discussed the convergence of compressed SGD using EF. However, unbiased relative compressors which satisfy (i) $\mathbb{E}_{\mathcal{C}}[\mathcal{C}(x)] = x$; and (ii) $\mathbb{E}_{\mathcal{C}}\|\mathcal{C}(x) - x\|^2 \leq \Omega\|x\|^2$ do not require EF. We compare the convergence of such unbiased compressors and absolute compressors with EF. With the notations above, [29] provide the following convergence result for unbiased compressors in the strongly convex case:

$$\mathbb{E}[f(\bar{x}_T)] - f^\star + \mu\mathbb{E}[\|x_T - x^*\|^2] \leq 64\Omega_n L(1+M/n)R_0 \exp\left[-\frac{\mu T}{4\Omega_n L(1+M/n)}\right] + 36\frac{(\Omega_n-1)D+\Omega\sigma^2/n}{\mu T},$$

Table 2: Summary of the benchmarks used

| Model | Task | Dataset | No. of Parameters | Optimizer |
|---|---|---|---|---|
| ResNet-18 [26] | Image classification | CIFAR-10 [35] | 11,173,962 | SGD+Nesterov momentum |
| LSTM [28] | Language modelling | Wikitext-2 [39] | 28,949,319 | Vanilla SGD |
| NCF [27] | Recommendation | Movielens-20M | 31,832,577 | ADAM [34] |

where $\Omega_n = \frac{\Omega-1}{n} + 1$. Comparing with Theorem 4, we find unbiased compressors have compression affecting the slower-decaying $\frac{1}{T}$ term. Although, we note that their convergence is in both the iterates and functional values, whereas ours is only in functional values.

## C   Addendum to numerical experiments

**Overview.** In this section, we provide:

i) The experimental settings and implementation details of our DNN experiments (§C.1).

ii) Further discussion on the large error-accumulation of Top-$k$ and its effect on total-error (§C.2).

iii) Logistic regression experiments (§C.3).

iv) Comparison against the state-of-the-art adaptive sparsifier ACCORDION [3]. (§C.4)

v) Experiment with Entire-model Top-$k$ (§C.5).

vi) Experiments without EF, and discussion on different forms of EF (§C.6).

### C.1   Experimental settings and implementation details

We implement the sparsifiers in `PyTorch`. For each method, a gradient reducer class is defined, which invokes the appropriate compression function and then perform the aggregation among the workers. Tables 2, 3, 4, and 5 provide the experimental details for each of the tasks. We used the default hyper-parameters provided in the mentioned repositories for each task.

Table 3: Image classification task

| | |
|---|---|
| Dataset | CIFAR-10 |
| Architecture | ResNet-18 |
| Repository | PowerSGD [57] |
| | See `https://github.com/epfml/powersgd` |
| License | MIT |
| Number of workers | 8 |
| Global Batch-size | $256 \times 8$ |
| Optimizer | SGD with Nesterov Momentum |
| Momentum | 0.9 |
| Post warmup LR | $0.1 \times 16$ |
| LR-decay | /10 at epoch 150 and 250 |
| LR-warmup | Linearly within 5 epochs, starting from 0.1 |
| Number of Epochs | 300 |
| Weight decay | $10^{-4}$ |
| Repetitions | 3, with different seeds |
| Hard-threshold: $\lambda$ values | $\{1.2 \times 10^{-2}, 7.2 \times 10^{-3}, 5 \times 10^{-3}, 3 \times 10^{-3}, 1.8 \times 10^{-3}\}$ |
| Top-$k$: $k$ values | $\{0.03\%, 0.06\%, 0.12\%, 0.3\%, 0.75\%\}$ |

### C.2   Top-$k$ suffers from large error accumulation

In Figure 4, we show the cascading effect (mentioned in §4.5) for the experiment in Figure 1. We observe that the error norm profile in Figure4 c closely follows the error compensated gradient norm profile in Figure4 b.

Table 4: Language modelling task

| Dataset | WikiText2 |
|---|---|
| Architecture | LSTM |
| Repository | PowerSGD [57] |
|  | See `https://github.com/epfml/powersgd` |
| License | MIT |
| Number of workers | 8 |
| Global Batch-size | $128 \times 8$ |
| Optimizer | vanilla SGD |
| Post warmup LR | $1.25 \times 16$ |
| LR-decay | /10 at epoch 60 and 80 |
| LR-warmup | Linearly within 5 epochs, starting from 1.25 |
| Number of Epochs | 90 |
| Weight decay | 0 |
| Repetitions | 3, with different seeds |
| Hard-threshold: $\lambda$ values | $\{4.5 \times 10^{-3}, 2.75 \times 10^{-3}, 1.6 \times 10^{-3}, 1.12 \times 10^{-3}\}$ |
| Top-$k$: $k$ values | $\{0.025\%, 0.05\%, 0.1\%, 0.2\%\}$ |

Table 5: Recommendation task

| Dataset | Movielens-20M |
|---|---|
| Architecture | NCF |
| Repository | NVIDIA Deep Learning Examples |
|  | See `https://github.com/NVIDIA/DeepLearningExamples` |
| Number of workers | 8 |
| Global Batch-size | $2^{20}$ |
| Optimizer | ADAM |
| ADAM $\beta_1$ | 0.25 |
| ADAM $\beta_2$ | 0.5 |
| ADAM LR | $4.5 \times 10^{-3}$ |
| Number of Epochs | 30 |
| Weight decay | 0 |
| Dropout | 0.5 |
| Repetitions | 3, with different seeds |
| Hard-threshold: $\lambda$ values | $\{2 \times 10^{-6}, 1.3 \times 10^{-6}, 1 \times 10^{-6}, 4 \times 10^{-7}\}$ |
| Top-$k$: $k$ values | $\{7.7\%, 9.5\%, 11.3\%, 13.7\%\}$ |
| License | Open Source |

In Figure 5 and Figure 6, we show that hard-threshold has a better convergence because of a smaller total-error in LSTM-WikiText2 and NCF-Ml-20m benchmarks. We note that we use the ADAM optimizer on the NCF-Ml-20m benchmark, and therefore our total-error insight is not theoretically justified in this case. Nevertheless, our experiment empirically confirms that the total-error perspective is useful for optimizers beyond vanilla SGD and momentum SGD.

### C.3 Logistic regression experiments

For the convex experiments, we consider the following $\ell_2$ regularized logistic regression experiment considered in [22][4]:

$$\min_{x \in \mathbb{R}^d} f(x) = \frac{1}{N} \sum_{i=1}^{N} \log(1 + \exp(-y_i \mathbf{A}[i,:]x)) + \frac{\mu}{2} \|x\|^2, \quad \text{where } \mathbf{A} \in \mathbb{R}^{N \times d}, y \in \mathbb{R}^N. \quad (31)$$

The function, $f(x)$ in (31) is $\mu$-strongly convex and $L$-smooth with $L = \mu + \frac{\lambda_{\max}(\mathbf{A}^T \mathbf{A})}{4N}$. As in [22], we use the step-size $\gamma = 1/L$, and $\mu = 10^{-4} \frac{\lambda_{\max}(\mathbf{A}^T \mathbf{A})}{4N}$. We use standard LIBSVM datasets [14],

---

[4]Open source code: `https://github.com/eduardgorbunov/ef_sigma_k`

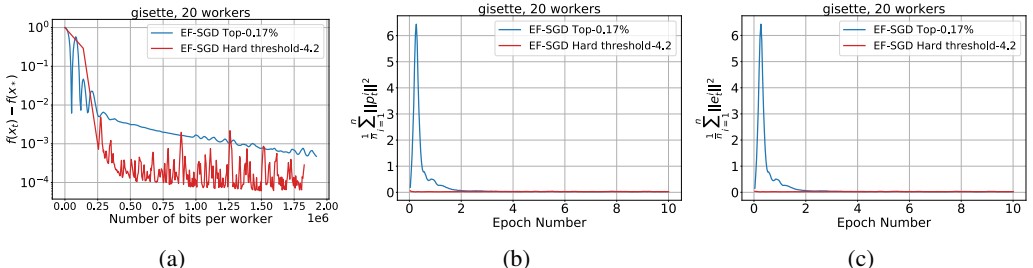

(a)          (b)          (c)

Figure 4: Convergence of Top-$k$ and Hard-threshold for a logistic regression model on `gisette` LIBSVM dataset with 20 workers: (a) Functional suboptimality vs. bits communicated; (b) Error-compensated gradient norm vs. Epoch; (c) Error-norm vs. iterations. Top-$k$ has large error-accumulation due to the cascading-effect.

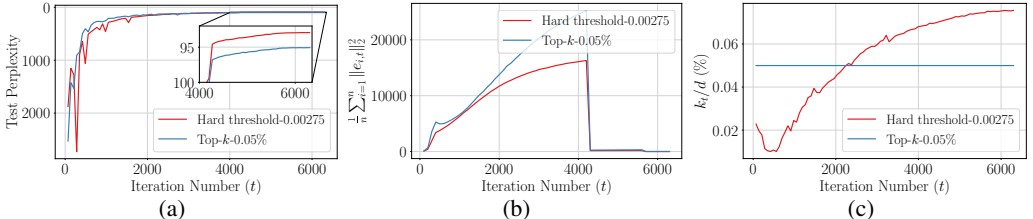

(a)          (b)          (c)

Figure 5: **Convergence of Top-$k$ and Hard-threshold for an LSTM on WikiText2 at $0.05\%$ average density:** (a) Test-perplexity vs. Iterations, (b) Error-norm vs. Iterations, (c) Density ($k_t/d$) vs. Iterations. $k = 0.05\%$ of $d$, and $\lambda = 0.0072$. Hard-threshold has better convergence than Top-$k$ because of a smaller total-error.

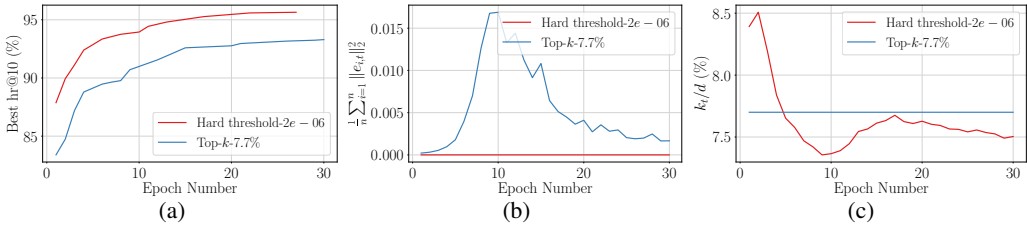

(a)          (b)          (c)

Figure 6: **Convergence of Top-$k$ and Hard-threshold for NCF on ML-20m at $7.7\%$ average density:** (a) Best Hit-rate@10 vs. Epochs, (b) Error-norm vs. Epochs, (c) Density ($k_t/d$) vs. Epochs. $k = 0.06\%$ of $d$, and $\lambda = 0.0072$. Hard-threshold has better convergence than Top-$k$ because of a smaller total-error.

and split the dataset into number of worker partitions. For distributed EF-SGD, we use a local batch size of 1 at each node, where the new batch is chosen uniformly at random at each step.

**Tuning the hard-threshold:** Our goal is to make $f(x_T) - f(x^\star) \leq \epsilon$, for a given precision, $\epsilon > 0$. We set $\lambda$ such that $d\gamma^2\lambda^2 = \epsilon$, i.e., $\lambda = \frac{\sqrt{\epsilon}}{d\sqrt{\gamma}}$.

*Justification:* Remark 5 states that by using a hard-threshold $\lambda > 0$, the noise due to compression is $d\gamma^2\lambda^2$. Due to this compression noise, we expect (although we did not prove) that $x_T$ will oscillate in a $d\gamma^2\lambda^2$ neighborhood of the optimum, $x^\star$, i.e. $\|x_T - x^\star\|^2 \leq d\gamma^2\lambda^2$. Furthermore, by $L$-smoothness, we have

$$f(x_T) - f(x^\star) \leq \frac{L}{2}\|x_T - x^\star\|^2.$$

Therefore, if we want to converge to a $\epsilon$-close functional-suboptimality value, $f(x_T) - f(x^\star)$, then ensuring $d\gamma^2\lambda^2 \leq \epsilon$ guarantees $\|x_T - x^\star\|^2 \leq \epsilon$, and implies, $f(x_T) - f(x^\star) \leq \frac{L}{2}\epsilon$. The above is an upper bound, and we observe in our experiments by using $\lambda = \frac{\sqrt{\epsilon}}{d\sqrt{\gamma}}$, gives $f(x_t) - f(x^\star) \leq \epsilon$.

### C.3.1 Extreme sparsification

In Figure 7, we perform extreme sparsification to train a logistic regression model on the `madelon` LIBSVM dataset. We compare Top-$k$ with $k = 1$, and hard-threshold with $\lambda = 14881$ set via $d\gamma^2\lambda^2 = 1.25 \times 10^{-4}$, so that they both communicate *same data volume*. In Figure 7b, we see

that Hard-threshold sparsifier does not communicate any elements in many iterations. Despite this, hard-threshold has faster convergence than Top-$k$ in Figure 7 a. Figure 7 c demonstrates that this is because hard-threshold has a smaller total-error than Top-$k$.

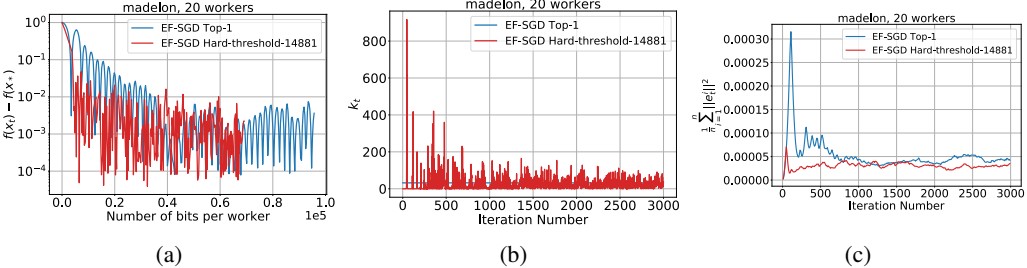

(a)                (b)                (c)

Figure 7: Convergence of Top-$k$ and Hard-threshold for a logistic regression model on `madelon` LIBSVM dataset with 20 workers: (a) Functional suboptimality vs. bits communicated; (b) parameters communicated vs. iterations; (c) error norm vs. iterations. Hard-threshold has a faster convergence than Top-$k$ even when it does not communicate any parameter in some iterations.

### C.3.2 Convergence to an arbitrary neighborhood of the optimum

For the experiments in this section, the uncompressed baseline is distributed gradient descent (GD). Unlike SGD, GD has linear convergence to the exact optimum. However, Distributed EF-GD does not converge to the exact optimum due to compression noise. To remedy this, Gorbunov et al. [22] introduced a family of variance-reduced compression algorithms that have linear convergence to the exact optimum. We consider algorithm `EF-GDstar` from [22] (known as `EC-GDstar` in [22]).

We empirically show that `EF-GDstar` with hard-threshold compressor, can converge to an arbitrarily small neighborhood around the optimum, for an appropriate choice of hard-threshold. Figure 8 and Figure 9 demonstrate the convergence of `EF-GDstar` using Hard-threshold and Top-$k$ sparsifiers with 20 workers and 100 workers, respectively. We choose (*i*) $k = 1$ for 20 workers and $k = 5$ for 100 workers, respectively; (*ii*) $\lambda = 2.98$, such that $d\gamma^2\lambda^2 = 5 \times 10^{-12}$. By using this $\lambda$, the compression error for hard-threshold is less than $5 \times 10^{-12}$ in Figures 8 c and 9 c. Moreover, hard-threshold converges to $f(x_T) - f(x^\star) \leq 5 \times 10^{-12}$ in both Figures 8 b and 9 b. Additionally, hard-threshold sends $1.7\times$ and $8\times$ less data than Top-$k$ in Figure 8 a and Figure 9 a, respectively. Furthermore, Figure 8 is an extreme sparsification scenario where hard-threshold communicates $< 1$ parameter per iteration per worker.

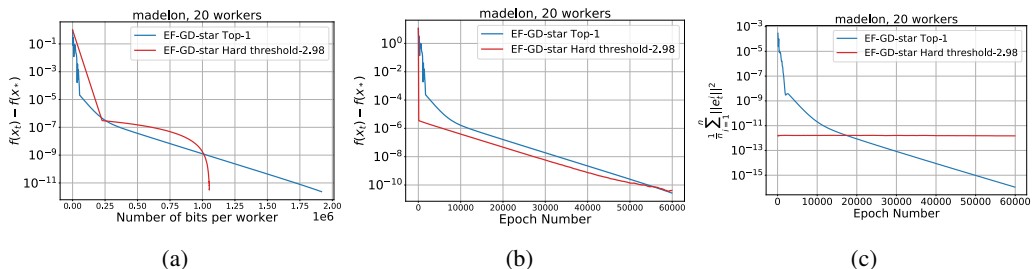

(a)                (b)                (c)

Figure 8: Convergence of `EF-GDstar` using Top-$k$ and Hard-threshold sparsifiers on a logistic regression model on `madelon` LIBSVM dataset with 20 workers: (a) Functional suboptimality vs. bits communicated; (b) functional suboptimality vs. epochs; (c) error-norm vs. epochs.

Our results demonstrate that it is possible to use the hard-threshold compressor to converge to an arbitrarily small neighborhood around the optimum. We leave the convergence analyses, and devising practical variants for future research.

### C.4 Comparison against ACCORDION

The experiment details are provided in 6, and the CIFAR-100 results are provided in Table 7.

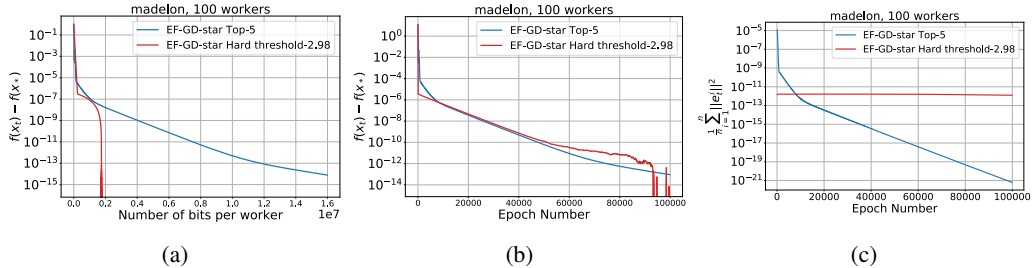

Figure 9: Convergence of `EF-GDstar` using Top-$k$ and Hard-threshold sparsifiers on a logistic regression model on `madelon` LIBSVM dataset with 100 workers: (a) Functional suboptimality vs. bits communicated; (b) functional suboptimality vs. epochs; (c) error norm vs. epochs.

Table 6: ACCORDION experiments

| | |
|---|---|
| Dataset | CIFAR-10 and CIFAR-100 |
| Architectures | ResNet-18 [26], SENet18 [30], GoogleNet [56] |
| Repository | PowerSGD [57] |
| | See `https://github.com/epfml/powersgd` |
| License | MIT |
| Number of workers | 8 |
| Global Batch-size | $256 \times 8$ |
| Optimizer | SGD with Nesterov Momentum |
| Momentum | 0.9 |
| Post warmup LR | $0.1 \times 16$ |
| LR-decay | $/10$ at epoch 150 and 250 |
| LR-warmup | Linearly within 5 epochs, starting from $0.1$ |
| Number of Epochs | 300 |
| Weight decay | $10^{-4}$ |
| Repetitions | 6, with different seeds |
| Accordion: $k_{\min}$ value | $0.1\%$ for both CIFAR-10 and CIFAR-100 |
| Accordion: $k_{\max}$ value | $1\%$ for CIFAR-10 and $2\%$ for CIFAR-100 |
| Hard-threshold: $\lambda$ values (Calculated using $\lambda = \frac{1}{2\sqrt{k_{\min}}}$) | ResNet-18-CIFAR-10: $4.73 \times 10^{-3}$ ResNet-18-CIFAR-100: $4.72 \times 10^{-3}$ GoogleNet-CIFAR-10: $6.37 \times 10^{-3}$ GoogleNet-CIFAR-100: $6.32 \times 10^{-3}$ SENet18-CIFAR-10: $4.68 \times 10^{-3}$ SENet18-CIFAR-100: $4.68 \times 10^{-3}$ |

Table 7: Comparison against ACCORDION [3] on CIFAR-100

| Network | Method | Accuracy (%) | Average Density (%) |
|---|---|---|---|
| ResNet-18 | Top-2% ($k_{\max}/d$) | 71.8 | 2.00 (1×) |
| | Top-0.1% ($k_{\min}/d$) | 70.6 | 0.10 (20×) |
| | ACCORDION | **71.6** | 0.57 (3.5×) |
| | Hard-threshold ($\frac{1}{2\sqrt{k_{\min}}}$) | 71.4 | **0.35 (5.7×)** |
| GoogleNet | Top-2% ($k_{\max}/d$) | 75.5 | 2.00 (1×) |
| | Top-0.1% ($k_{\min}/d$) | 73.1 | 0.10 (20×) |
| | ACCORDION | 74.2 | 0.48 (4.2×) |
| | Hard-threshold ($\frac{1}{2\sqrt{k_{\min}}}$) | **75.0** | **0.38 (5.3×)** |
| SENet18 | Top-2% ($k_{\max}/d$) | 71.9 | 2.00 (1×) |
| | Top-0.1% ($k_{\min}/d$) | 70.1 | 0.10 (20×) |
| | ACCORDION | 71.0 | 0.55 (3.6×) |
| | Hard-threshold ($\frac{1}{2\sqrt{k_{\min}}}$) | **72.1** | **0.36 (5.6×)** |

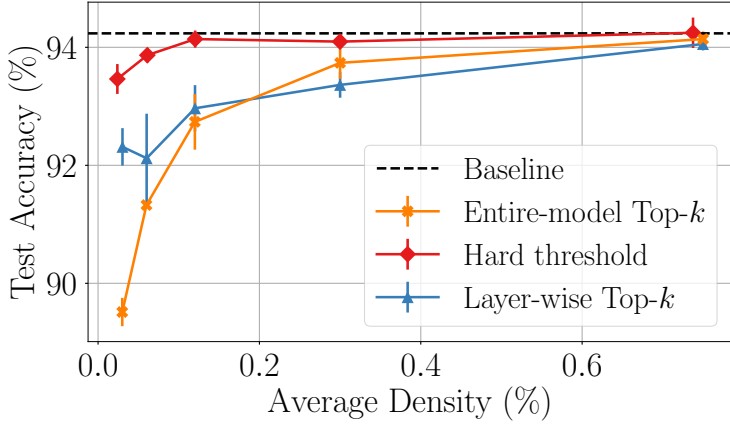

Figure 10: ResNet-18 on CIFAR-10

Figure 11: **Test metric vs. Data volume for entire-model compression**. The dashed black line in each plot denotes the no compression baseline. Each setting is repeated with three seeds, and we plot the average with standard deviation. For description on parameters, see Tables 3, 4, and 5.

## C.5  Entire-model sparsification

Sparsification can be performed in two ways: layer-wise or entire-model. In layer-wise sparsification, the sparsifier is invoked individually on each tensor resulting from each layer. In contrast, in entire-model sparsification, the sparsifier is applied to a single concatenated tensor resulting from all layers. Since hard-threshold is an element-wise sparsifier, layer-wise and entire-model sparsification result in the same sparsified vector. However, it is expected that layer-wise and entire model vary substantially for Top-$k$. Layer-wise Top-$k$ is used in all practical implementations [47, 37, 62] because performing entire-model Top-$k$ is both compute and memory intensive.

While we employ layer-wise Top-$k$ in our experiments, we present in Figure 11 the *test metric vs. data volume* experiment for ResNet-18-CIFAR-10 benchmark (Figure 2a) using entire-model Top-$k$. We find that hard-threshold is more communication-efficient than entire-model Top-$k$ as well. Notably, at an average density ratio of $0.003\%$, hard-threshold has more than $4\%$ higher accuracy than entire-model Top-$k$.

## C.6  Error-Feedback (EF)

In this section, we discuss various aspects of EF (or memory). Particularly, in §C.6.1 we investigate if hard-threshold is more communication-efficient than Top-$k$ without EF. Then, in Section C.6.2, we discuss and compare the different ways to perform EF in the literature.

### C.6.1  Convergence without EF

To understand how the sparsifiers perform without the EF, we conduct experiments without EF for ResNet-18 benchmark. We report this in Figure 12. Similar to the with EF case, we find that hard-threshold has better convergence than Top-$k$. We note that with EF, hard-threshold achieved baseline performance at an extreme average density of $0.12\%$. However, without EF, hard-threshold fails to achieve baseline performance ($94.2\%$) even at a significantly higher average density of $5\%$. Hence, EF is a necessary tool to ensure faster convergence.

### C.6.2  Different types of EF

For optimizers other than vanilla SGD, one can compress and aggregate quantities other than stochastic gradients (such as momentum). Consider an example for SGD with Nesterov momentum, where the compression and aggregation can be performed in the following two ways:

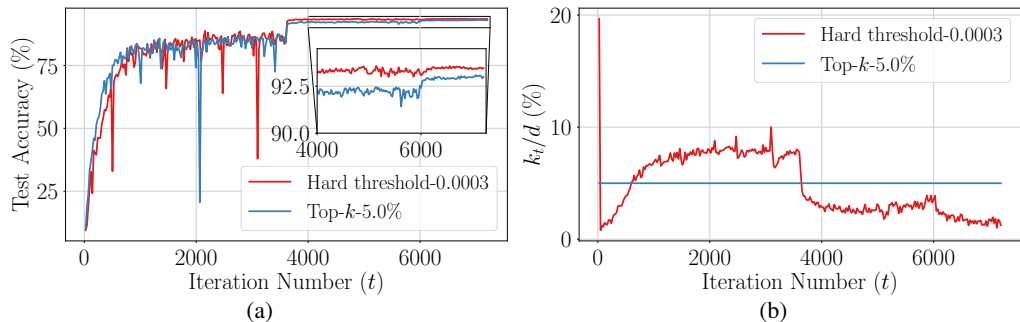

(a)

(b)

Figure 12: Top-$k$ and hard-threshold without error compensation for ResNet-18 on CIFAR-10: (a) Accuracy vs. Iterations, (b) density, $(k_t/d)$ vs. iterations. Average density is $5\%$ for Top-$k$ and $4.7\%$ for hard-threshold.

---

**Algorithm 2:** Distributed EF SGD with momentum by using gradient compression

**for** *worker* $w = 1, .., W$ **in parallel do**
  **for** *iteration* $t = 1, 2, \cdots$ , **do**
    Compute local stochastic gradient $g_w$
    $\Delta_w \leftarrow g_w + e_w$
    $\mathcal{C}(\Delta_w) \leftarrow \texttt{COMPRESS}(\Delta_w)$
    $e_w \leftarrow \Delta_w - \texttt{DECOMPRESS}(\Delta_w)$
    $\mathcal{C}(\Delta) \leftarrow$
      $\texttt{AGGREGATE}(\mathcal{C}(\Delta_1), \ldots, \mathcal{C}(\Delta_W))$
    $\Delta' \leftarrow \texttt{DECOMPRESS}(\mathcal{C}(\Delta))$
    $m \leftarrow \lambda m + \Delta'$
    $x \leftarrow x - \gamma(\Delta' + m)$

**Algorithm 3:** Distributed EF SGD with momentum by using update compression

**for** *worker* $w = 1, .., W$ **in parallel do**
  **for** *iteration* $t = 1, 2, \cdots$ , **do**
    Compute local stochastic gradient $g_w$
    $m_w \leftarrow \lambda m_w + g_w$
    $u_w \leftarrow m_w + g_w$
    $\Delta_w \leftarrow u_w + e_w$
    $\mathcal{C}(\Delta_w) \leftarrow \texttt{COMPRESS}(\Delta_w)$
    $e_w \leftarrow \Delta_w - \texttt{DECOMPRESS}(\Delta_w)$
    $\mathcal{C}(\Delta) \leftarrow$
      $\texttt{AGGREGATE}(\mathcal{C}(\Delta_1), \ldots, \mathcal{C}(\Delta_W))$
    $\Delta' \leftarrow \texttt{DECOMPRESS}(\mathcal{C}(\Delta))$
    $x \leftarrow x - \gamma(\Delta')$

---

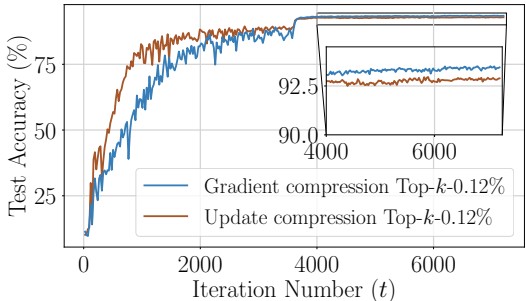

Figure 13: Test Accuracy for gradient compression vs. update compression for Top-$k$ on ResNet-18 on CIFAR-10. We experiment with three different seeds, and the plot represents the run with highest final accuracy for each setting. The test accuracy statistics $(\mu \pm \sigma)$ are: Gradient compression $(92.96 \pm 0.39\%)$ and update compression $(90.78 \pm 2.03\%)$.

- **Gradient compression.** This was proposed in [57] and is depicted in Algorithm 2. In the case of SGD with Nesterov momentum, this update rule ensures that every worker maintains the same momentum state. However, the updates to momentum is sparse, as the momentum is calculated using sparsified gradients.

- **Update compression.** This was proposed in [37], and is depicted in Algorithm 3. In the case of SGD with Nesterov momentum, every worker maintains a different momentum state calculated from their local stochastic gradients. Although updates to the momentum state is dense in this case, the momentum state is completely unaware of the compression and does not reflect the actual history of the updates. In order to circumvent this issue, Lin et. al. [37] had proposed momentum factor-masking to clear old local momentum states of a parameter once the parameter is updated. However, it is not easy to devise such modifications for optimizers which maintain multiple states derived from complicated calculations, such as RMSProp and ADAM.

**Nomenclature for Algorithm 2 and 3.** In Algorithm 2 and 3 we show the distributed training loop. We denote the learning rate by $\gamma$, momentum factor by $\lambda$, the model parameters by $x \in \mathbb{R}^d$, the momentum at worker $w$ by $m_w$, and the error at worker $w$ by $e_w$. At the beginning of the training, $m_w$ and $e_w$ are initialized to zero for all workers. By COMPRESS, DECOMPRESS, and AGGREGATE we denote the compression, decompression, and aggregate function, respectively.

We also conduct experiments for Top-$k$ on ResNet-18 benchmark by using aforementioned update rules and find that gradient compression (Algorithm 2) results in better performance (see Figure 13). In light of the above discussion and experimental evidence, we stick to gradient compression (Algorithm 2) for our main experiments.

## D  How to tune the hard-threshold?

Substituting $v^2 = d\lambda^2$ for hard-threshold in Theorem 6 we get

$$\frac{1}{T} \sum_{t=0}^{T-1} \mathbb{E}\|\nabla f(x_t)\|^2 \leq \frac{4(f(x_0)-f^\star)}{\gamma T} + \frac{2\gamma L(M\zeta^2+\sigma^2)}{n} + 2\gamma^2 L^2 d\lambda^2. \tag{32}$$

Similarly, substituting $\delta = \frac{k}{d}$ for Top-$k$ in Theorem 7 we get

$$\frac{1}{T} \sum_{t=0}^{T-1} \mathbb{E}\|\nabla f(x_t)\|^2 \leq \frac{8(f(x_0)-f^\star)}{\gamma T} + \frac{4\gamma L(M\zeta^2+\sigma^2)}{n} + \frac{8\gamma^2 L^2 d}{k}\left(\left(\frac{2d}{k} + M\right)\zeta^2 + \sigma^2\right). \tag{33}$$

We ignore the first two terms unaffected by compression in (32) and (33), and focus on the last term. To ensure that hard-threshold has better convergence than Top-$k$ we have

$$2L^2 d\lambda^2 \leq \frac{8L^2 d}{k}\left(\left(\frac{2d}{k} + M\right)\zeta^2 + \sigma^2\right),$$

that is,

$$\lambda \leq \frac{2}{\sqrt{k}}\sqrt{\left(\frac{2d}{k} + M\right)\zeta^2 + \sigma^2}.$$

Therefore, if $\hat{M}$, $\hat{\zeta}$, and $\hat{\sigma}$ are estimates of $M$, $\zeta$, and $\sigma$, respectively, then we suggest setting the threshold as

$$\lambda \sim \frac{2}{\sqrt{k}}\sqrt{\left(\frac{2d}{k} + \hat{M}\right)\hat{\zeta}^2 + \hat{\sigma}^2}.$$

In our comparison against ACCORDION, we assume $\hat{\zeta} = 0$ (homogeneous distributed data), and $\hat{\sigma} \sim \frac{1}{4}$. This leads us to the hard-threshold value

$$\lambda \sim \frac{1}{2\sqrt{k}}.$$

We find that $\lambda = \frac{1}{2\sqrt{k_{\min}}}$ has better performance (with similar total-data volume) than Top-$k_{\min}$ in Tables 1 and 7.