# OpenReview forum: "Rethinking gradient sparsification as total error minimization"
_NeurIPS.cc/2021/Conference — NeurIPS 2021 Spotlight_

### Official Review · Reviewer_o9PJ · 2021-07-02

**Rating:** 7
**Confidence:** 3

**Summary:**

The paper considers gradient sparsification for learning in distributed setup, and advocates using a hard-threshold sparsifier combined with error-feedback mechanism. The paper shows that such algorithm is _optimal_ in a certain sense, and give several convergence guarantees for the error-feedback algorithms using absolute compressors and relative compressors. The empirical performance of HT and top-k compressors are also compared.

**Limitations And Societal Impact:**

I think authors should additionally mention when can such variable-communication-load methods may not useful or usable.

**Main Review:**

Post-rebuttal: Thank you for the reply. My concerns have been addressed well. Raising the score to 7.

---

__TL;DR.__ I think this paper presents several meaningful theoretical results as a contribution. However, I think some of the paper's claims are being quite oversold.

__Strengths.__ Some of the theoretical results are definitely very cool to have. I believe that the convergence results in Section 5 (Theorems 2--5) is a nice contribution, and would be of interest to the distributed learning society, especially to those who study error-feedback mechanisms. Also, the proof technique going through the perturbed iteration analysis (via Lemma 10) is quite neat. Finally, the manuscript seems to discuss the related work relatively well.

__Claims on optimality.__ I am very worried about the paper's claims about the optimality. The word "optimal" is a very bold, and should be used with a great care (in my opinion), as they could be quite misleading without delivering the assumptions and conditions it relies on. For instance, the abstract states that top-k is "communication-optimal given a per-iteration $k$-element budget." But, what exactly does the word "communication-optimal" here mean? It is very easy to understand the statement as saying that such algorithm gives the hypothesis with a smallest loss---either in expected or high-probability sense. If I understood correctly, I think the paper is pointing to lemma 2, where authors state that Top-k gives the sparsified version of the error-feedback (or actually any signal) that has the smallest squared distortion from the original gradient signal. This also relies on the assumption that the choice of sparsification does not affect the subsequent gradient signals. This discrepancy gets more significant for the claims on the optimality of hard-threshold methods, where this "independency assumption" is critical for the proof. I believe that these ill-specified claims on the (possibly vacuous notions of) optimality should either be toned down to a certain degree to help readers better understand what the paper is contributing. Also, I think presenting the optimality claims in a form of lemmata without proofs is unnatural, no matter how straightforward the proofs are. I recommend either stating the claims in plain words without a formalization, or provide the result-specific assumptions clearly in the lemma statement---so that it is self-contained---and give at least a formal proof.

__Stating the limitations.___ If I understood correctly, such hard-threshold sparsifiers may need a very high communication throughput at some epochs (mostly earlier). However, there are many setups such high peak communication rate is undesirable, due to the limited capacity of the communication channel (but when the delay is crucial). In such cases, having a constant communication rate could be beneficial. I think authors should discuss such scenarios to appropriately deliver the cases where the considered hard-threshold methods are desirable, and the cases they are not (sorry if I missed these parts).

__Clarity: Why error-feedback?__ I am not entirely sure what is the big motivation behind considering the error-feedback mechanisms for this paper. Line 51 says: "Consequently, we consider sparsification using the error-feedback mechanism, ..." but I couldn't really locate the part that necessitates considering error-feedback. Could you please further clarify?

__Clarity: $\gamma_t$.__ The quantity $\gamma_t$ appears at line 133, but I do not think this quantity is defined or introduced properly in the text.

__Clarity: Assumption 2.__ I do not think Assumption 2 is explicitly assumed in any theorem appearing in the main text. But if I am correct, it is implicitly used for every theorems that use $R_0 := \lVert x_T - x^\star \rVert$ in its bounds.

__Question: $v$-dependency__ It was unexpected to me that the convergence guarantees for the absolute compressors $C_v ( \cdot ) $ does not depend explicitly on the compression factor $v$, especially considering the fact that the guarantees for the $\delta$-contraction operators contain a term that is inversely proportional to $\delta$ (see Theorem 2 & Remark 5, for instance). Any further discussion explaining why such discrepancy happens (especially for the case $\delta \to 0, v \to \infty$) would be very nice to have; does this suggest the existence of a tighter bound for $\delta$-contraction operators under additional assumptions the size of $p_{i,t}$?

__Suggestion: Discussing $P_T$.__ While the quantity $P_T$ appears in many optimization literature, the meaning of the quantity may not be very straightforward for the readers who are relatively new to the field (like myself), especially because it gives the performance bound for $\bar{x}_T$ generated by some weights $w_t$. Giving more ideas about the quantity may help the readers a lot, including whether one can choose $w_t$ arbitrarily or not, whether it implies that we should use an averaging scheme...

**Time Spent Reviewing:**

20

---

> ### Author Response · Authors · 2021-08-10
> **Author Response to Reviewer o9PJ**
>
> We are grateful to the reviewer for the constructive feedback and for providing a positive assessment of our paper. The reviewer has raised some valid questions and provided many mindful suggestions. The following are our responses to the reviewer’s comments:
>
> **Claims on optimality:** We thank the reviewer for this comment. Our use of the word *optimal* is motivated by recent literature in compressed distributed optimization [ 39, 18, 13, 4] that focus on the compression-error (see Footnote 2). Any compressed optimization convergence analysis captures the effect of compression via the compression-error, and this effect is always inverse---the lower the compression error, the better the optimization upper bound is. Please refer to one-step descent Lemmas 12 and 15. To derive a convergence rate from these lemmas, we need to use the worst-case compression error/factor . For instance, to derive a convergence rate for $\delta$-contraction operators in Remark 5 and Remark 7, one uses the worst-case compression factor in equation (7) and lines 211-212. Due to this, [39, 18, 13, 4] directly optimize for this worst-case compression factor and call their compressors as *optimal*.
>
> However, for gradient sparsification with a fixed $k$ element communication per iteration, this worst-case bound is not insightful. For example, to sparsify a $d$-dimensional vector, both Random-$k$, and Top-$k$ have the same compression factor, $k/d$, although Top-$k$ performs significantly better in practice than Random-$k$; please see further discussion in [8]. But we know that for a given signal, Top-$k$ attains the lowest compression-error among all sparsifiers with $k$ element communication budget. And therefore, we state Top-$k$ as the communication-optimal sparsifier under a fixed $k$-element communication budget. Precisely, this is the message behind Lemma 2 as the reviewer has correctly identified. We will elaborate on this in detail in the final version.
>
> Next, we will mitigate the confusion regarding Lemma 3. We agree that we should provide proof for Lemma 3, and will do so in the Appendix of the revised paper. Please note that we have been careful to stress throughout the text that hard-threshold is communication optimal *in our communication complexity model*, and have stated that the total-error cannot be directly minimized (Lines 175-177). We coined the term *total-error* because this term captures the compression error in the entire training process. We formalize our communication complexity model in (5) after simplifying the total error; please see Lines 191-193.
>
> **Stating the limitations.** We thank the reviewer for pointing this out and we agree with the comment. Indeed there are scenarios where a predetermined compression ratio for an iteration is desirable. Examples include dynamic network environments such as a public cloud or a shared cluster with colocated jobs [Abdelmoniem et al., 2021]. In such a setting, one may want to adjust the compression knobs according to the current network bandwidth so that training finishes within a time budget, and therefore hard-threshold is not a good candidate for this setting.
>
> In a standard distributed cluster setting with a dedicated network, if communication is a bottleneck for Top-$k$, i.e., there is not a complete overlap between communication and computation, so a hard-threshold with the same total communication volume will have non-overlapped communication in the iterations with high data transmission, but may also have completely overlapped communication in iterations with low data transmission. Thus, hard-threshold can have less non-overlapped communication time than Top-$k$ in this case. This happens in large-scale CRT models such as DeepLight which has 90% of non-overlapping communication. Considering the opposite, if there is complete overlap between computation and communication for Top-$k$, then a hard-threshold with the same total communication volume may have non-overlapped communication time in some iterations with high data transmission. Here, we ignored two important aspects of hard-threshold: (i) Hard-threshold has better statistical efficiency than Top-$k$, thus one may require smaller iterations to a target accuracy. (ii) Hard-threshold has negligible compression overhead in comparison to Top-$k$ (lines 73-79).
>
> [Abdelmoniem et al., 2021] DC2: Delay-aware compression control for distributed machine learning. IEEE INFOCOMM 2021.
>
> **Why error-feedback?** This is a critical question and we thank the reviewer for asking this. Please allow us to discuss the development of the error-feedback theory in this context. Error-feedback was first empirically introduced by Seide et al. [40] in 2014, to alleviate the convergence of 1-bit low-precision SGD in training language models. But, for the next 5 years, the community was unaware about why error-feedback is an important technique. In 2018, Stich et al. [45] were the first to theoretically establish the convergence of SGD using $\delta$-contraction operators with error-feedback, which was extended to the distributed setting by Zheng et al. in [55]. More interestingly, in subsequent work, Karimireddy et al. [29] theoretically showed that error-feedback can remedy the convergence issues of aggressive quantizers, such as 1-bit/Sign SGD, as well as biased $\delta$-sparsifiers, such as Top-$k$, Random-$k$, etc. We also refer to [46] for a detailed discussion on the error-feedback framework. In a nutshell, without error-feedback, most sparsifiers (which also belong to the class of $\delta$-contraction operators) diverge [29,46]. Moreover, the best compression ratios for gradient sparsification are achieved when we use error-feedback. Please refer to Table 1 in the comprehensive survey [54], where all implemented sparsifiers use error-feedback. This makes error-feedback an "indispensable and essential" technique for gradient sparsification. And this was the "big motivation" of using error-feedback in our work as it is focussed on gradient sparsification. However, we also thank the reviewer for pointing out "Line 51 says: "Consequently, we consider sparsification using the error-feedback mechanism, ...."." We will rewrite this line to justify why we use the error-feedback.
>
> **Clarity:$\gamma_t$**. Thank you for this important observation. We will clearly mention that $\gamma_t>0$ is the stepsize sequence.
> *Assumption 2.* We thank the reviewer for catching this typo and rectifying us. We will mention Assumption 2 in the theorems where it is supposed to appear.
>
> **$\upsilon$-dependency.** We apologize for the confusion. Absolute compressors’ convergence is in terms of $\kappa$; please see lines 244-250. We will make this clearer by directly compressing the error-compensated gradients in Algorithm 1 instead of the error-compensated updates as done presently.
>
> **Discussing $P_T$.** The quantity $P_T$ in our paper denotes the expected suboptimality gap, $E[f(\bar{x}_T)]-f^\star$ at averaged iterate, $\bar{x}_T$, where $f^*$ is the global minimum, defined in Assumption 2. We understand and appreciate the reviewer's concern in clarifying the meaning behind this quantity. Here $w_t$ is a carefully chosen set of weights such that we achieve the convergence result. We will highlight its meaning with references at the beginning of Section 5.1 where it appears for the first time.

---

### Official Review · Reviewer_eex4 · 2021-07-16

**Rating:** 7
**Confidence:** 3

**Summary:**

In this paper, the authors demonstrate that, in the context of distributed optimization problems with $n$ workers, the hard-threshold sparsifier is the optimal sparsifier for a proposed communication complexity model which where the goal is to minimize the total error for a sequence of responses. This allows the authors to compare the sum of compression errors for various algorithms and to demonstrate that while for the per-iteration $k$-element budget the Top-$k$ sparsifier is optimal, when it comes to total error the hard threshold sparsifier is better.

The authors also compare the convergence rates of the top-$k$ sparsifier vs the hard threshold sparsifer for image classification, language modeling, and recommendation tasks.

**Limitations And Societal Impact:**

This is a theoretical paper and has limited societal impact.

**Main Review:**

I find this paper interesting and vote to accept it. I think it is an interesting result but unfortunately I am not an expert in the field and so am not sure about it's significance in the context of the field. I do like the communication complexity model.

**Time Spent Reviewing:**

2

---

> ### Author Response · Authors · 2021-08-10
> **Author Response to Reviewer eex4**
>
> We thank the reviewer for the effort in reviewing our paper. We also thank the reviewer for providing a positive assessment of our work and for appreciating our communication-complexity model which is indeed a key contribution of the paper.

---

### Official Review · Reviewer_VFA3 · 2021-07-16

**Rating:** 7
**Confidence:** 5

**Summary:**

The paper studies the role of compression operators on the convergence of Distributed SGD with Error Feedback. In particular, via simple observations, the authors conclude that a hard-threshold sparsifier with a carefully tuned threshold parameter minimizes the total error appearing in the analysis because of the presence of compression. Moreover, they show empirically the connection between poor behavior of EF-SGD with Top-k compression and the severe error accumulation.

Motivated by these observations, the authors derive new convergence guarantees for EF-SGD with absolute compressors. This class of compressors covers hard-threshold sparsifier. The derived bounds show that the compression does not affect the slowest terms in the bound. Moreover, the authors derived the first complexity result in the non-convex case for EF-SGD with $\delta$-cnotraction operators without bounded gradient assumption and $n > 1$ (though, under bounded data dissimilarity). Finally, the paper contains a good empirical study of the performance of EF-SGD with the hard-threshold sparsifier. The authors also provide an insight on how to tune the threshold parameter in order to outperform EF-SGD with the Top-k operator.

**Limitations And Societal Impact:**

The authors adequately addressed the limitations and potential negative societal impact of their work.

**Main Review:**

## Strengths

1. **Simple but important observations about total error minimization.** The paper provides a closer look at the convergence of EF-SGD and identifies what quantity should be minimized in order to get better results. That is, via the sequence of simple observations, authors show that a hard-thresholding sparsifier (with fine-tuned threshold parameter) is the optimal choice in terms of the total error minimization. Moreover, the authors properly explain all the details and support their theoretical observations with empirical findings (e.g., see Figures 1 and 3).

2. **Clarity and proofs.** The paper is clearly written. The proofs are easy to follow and contain only a couple of typos.

3. **New results for EF-SGD.** The authors derived new results for the convergence of EF-SGD with absolute compression for strongly convex, convex, and non-convex objectives. The slowest terms in the derived bounds have a linear speedup and are not affected by compression-dependent parameters. This is a good property since the derived bounds match the ones for SGD without compression if the target accuracy is small enough / the number of communication rounds is large enough. Moreover, the authors derived the first convergence result for EF-SGD for $\delta$-contraction operators without assuming boundedness of the gradients, but under Assumption 4, that bounds dissimilarity between local loss functions. Although the proofs substantially rely on the known techniques, the obtained results are quite good.

4. **Numerical experiments** show a connection between the behavior of EF-SGD with Top-k and hard-thresholding sparsifier and "error buildup". Therefore, these numerical results justify the insights provided in Section 4.

## Weaknesses

1. **No analysis for the arbitrary heterogeneous case for non-convex objectives (minor).** The derived bounds in the non-convex case substantially rely on Assumption 4 that bounds the dissimilarity between local loss functions. Although this is a significant limitation, previous works on EF-SGD rely on even stronger assumptions. Therefore, this is a minor drawback.

## Questions and Comments

1. **Rates for EF-SGD with $\delta$-contraction operator.** The rates shown in Remarks 5 and 7 can be significantly improved via the results from [19]. Although, in [19], Assumption 3 is not considered it can be easily cast in the general framework from [19] via the following derivation: $\frac{M}{n}\sum_{i=1}^n\|\nabla f_i(x^k)\|^2 + \sigma^2 \leq 2LM(f(x^k) - f(x^*)) + \frac{M}{n}\sum_{i=1}^n \|\nabla f_i(x^*)\|^2 + \sigma^2$. Using the results from [19] one can actually show the linear speedup even for $\delta$-contraction operators in the $\mathcal{O}(1/T)$ and $\mathcal{O}(1/\sqrt{T})$ decaying terms for $\mu > 0$ and $\mu = 0$ respectively. Therefore, the conclusion from lines 270-272 is not correct.

2. **equation after line 654:** the enumerator in the second term should be $\mu L^2(1+M/n)^2R_0 + L\kappa^2 \ln(T)$.

3. **line 247, $\nu = \gamma_t \kappa$:** It is better not to use $\kappa$ in the definition of $\nu$ because $\kappa$ usually denotes the condition number of the problem in the optimization literature.

4. **lines 283-284:** This is done for $\delta$-contraction operators in [19] and in Qian, X., Dong, H., Richtárik, P., & Zhang, T. Error Compensated Loopless SVRG for Distributed Optimization, Qian, X., Richtárik, P., & Zhang, T. (2020). Error compensated distributed SGD can be accelerated. arXiv preprint arXiv:2010.00091.

5. **Lemmas 6 and 7.** First of all, one should add that $\gamma = \min\left(\frac{1}{d}, \sqrt{\frac{r_0}{cT}}\right)$. Next, Lemma 7 can be tightened when $c$ is small, see Lemma D.3 from [19].

6. **Lemma 9.** When $c = 0$ this result is incorrect since the logarithmic factor becomes infinity. See Lemma D.2 from [19] for the correct version.

7. **Lemma 11.** It is better to cite the assumptions of the lemma in the statement (or at the beginning of the subsection).

8. **lines 623-627:** The discussion in these lines should be rewritten after applying the corrections suggested in comment 1. Moreover, one should also say that $\lambda$ can be large.

## Comment after rebuttal
I thank the authors for their response. I have read other reviews as well. Overall, my evaluation of the work remains the same. Therefore, I recommend the paper for acceptance and hope that the authors will apply all necessary corrections mentioned in the reviews.

**Time Spent Reviewing:**

6 hours

---

> ### Author Response · Authors · 2021-08-10
> **Author Response to Reviewer VFA3**
>
> We thank the reviewer for the positive review of our paper. We are glad that the reviewer considers total error minimization as an important contribution to our paper. Indeed, this is the heart of our paper. Below we discuss the questions and the comments of the reviewer.
>
> **No analysis for the arbitrary heterogeneous case for non-convex objectives (minor).** We sincerely thank the reviewer for pointing out this interesting aspect. We will address the arbitrary heterogeneous case in our future work.
>
> ## Questions/Comments
>
> **1. Rates for EF-SGD with $\delta$-contraction operator.** We sincerely thank the reviewer for bringing the tighter rates in [19] to our notice. We will rewrite this discussion accordingly.
>
> **2. Typo after line 654.** We thank the reviewer for indicating this typo. We will correct this.
>
> **3. Notation $\kappa$.** We thank the reviewer for this comment; and yes, you are right. We will use a better notation in the final version of the paper.
>
> **4. Related works.** We thank the reviewer for pointing out these works. We will indeed mention them and include more discussions in a proper context in the final version of the paper.
>
> **5. Lemmas 6 and 7.** We thank the reviewer for giving us this insight. We will use the tighter result from [19].
>
> **6. Lemma 9.** We thank the reviewer for spotting this. We will correct it.
>
> **7. Assumption in Lemma 11.** We thank the reviewer for this insightful comment. Indeed we will state the assumptions in the main statement of Lemma 11.
>
> **8. Lines 623-627.** As mentioned in comment 5, we will now compare absolute compressors and $\delta$-contraction operators using [19]. We will also state that $\lambda$ can be arbitrarily large.

---

### Official Review · Reviewer_TZ2h · 2021-07-16

**Rating:** 7
**Confidence:** 4

**Summary:**

This paper analyzes the hard-threshold sparsifier in distributed SGD with convergence analysis and extensive experiments. Main contributions include: 1. Provides upper bounds of the optimization errors for the hard-threshold sparsified distributed SGD, which improves from top-k compressor with linear speedup and compressor operator parameter dependence. 2. Conducts extensive experiments to demonstrate the benefits of hard-threshold sparsifier, achieving much better performance given the same average compression density (# of used coordinates / # of coordinates of DNN weights).

**Limitations And Societal Impact:**

Yes.

**Main Review:**

1. I am trying to understand why the proposed communication-complexity model is new, it seems to me it is an adaptive compression operator allowing different # of coordinates to be sent for each iteration. And the section 4.4 and Lemma 3 seems incorrect to me, since the optimization over B is sequential, the choice of first block will change the second block of A, so I didn’t quite follow why the proposed sparisfier is optimal for this communication-complexity model.
2. On the confusion of total error minimization. In the paragraph starting from line 44, it seems to the authors are trying to present a new perspective that is not minimizing per iteration compression error but total compression errors, but the last sentence seems to me conveying the message of considering fixed total communication budget, which is exactly what I think other papers have discussed, how to tradeoff optimization error with total communication budgets. So there may be some confusion in this paragraph.
3. The authors provide upper bounds of optimization errors for the proposed spasifier, in strongly convex, convex, and nonconvex settings respectively. Those upper bounds improve from the results using top-k sparsifier in terms of linear speedup and compressor parameter dependence.
4. The provided upper bounds, however, seem to me not clearly they are more communication-efficient than the top-k sparsifier, since there is no characterization of the total number of coordinates being used in theory. Ideally, if we set the threshold be very small, most coordinates will be used. In experiments, the authors provide solid results showing that the hard-threshold sparsifier does use less coordinates in total than the top-k.
5. The total error minimization perspective seems to me more like an observation of the consequence using a hard-threshold sparsifier, it may need further arguments to show it is the reason for better performance.
6. The paper is in general well written and very clear, the extensive experiments are helpful for the understanding of the practical benefits of the hard-threshold sparsifier.

**Time Spent Reviewing:**

10

---

> ### Author Response · Authors · 2021-08-10
> **Author Response to Reviewer TZ2h**
>
> We sincerely thank the reviewer for the positive assessment of our paper and for constructive feedback. Below we address the points mentioned by the reviewer:
>
>
> **1.Communication-complexity model and total error minimization.** We thank the reviewer for pointing this out. We accept that there is a slight misunderstanding about the simplification in our communication-complexity model. As mentioned in line 192, we assume a simplified model---Instead of the error-corrected update, $\gamma g_t +e_t$, we consider a sequence of *fixed* vectors $(a_t)_{t \in T}$  and formalized the optimization problems (5) and (6) in Section 4.4. Hence as the reviewer asserted, the optimization in Section 4.4 is not sequential. We will clarify this.
>
> We note that the existing communication-optimal strategies [18, 38, 13, 4] minimize the compression factor (see Footnote 2) under a budget for each vector; please see lines 221-227. In contrast, the novelty of our communication-complexity model is that it better captures the total error. We owe this insight to Theorem 1 that accounts for the effect of sparsification in the entire training process. Please refer to the statement of Theorem 1, which presents the convergence of distributed error-feedback SGD with compressed communication (here, sparsification). The third term on the right hand side of the inequality captures the effect of compression between the error-corrected update, $\gamma g_t +e_t$, and its compressed form, $C(\gamma g_t +e_t)$ over all iterations, $t=0,1,\cdots, T-1$. This is a well-accepted theoretical result and appears abundantly in the literature; please see [29,46]. Motivated by this result, we consider the total error perspective as a communication complexity model----where simplified total error is minimized under a total communication budget. Hard-threshold sparsifier comes out as the communication-optimal compressor under this communication complexity model. Therefore, the total-error minimization is not the consequence of using the hard-threshold sparsifier; it is the reason behind the hard-threshold as a communication-optimal sparsifier. Moreover, we substantiate this with insights from our experiments in Figures 1, 3, 5, and 6.
>
> Next, we respectfully note that we use the overall communication budget to denote the total communication throughout the training.
>
>
> **2. Convergence of hard-threshold.** We thank the reviewer for this question. The reviewer is correct---Our convergence results do not show that the hard-threshold is more communication-efficient than Top-$k$. Because we do not characterize the average data transmission for a threshold. However, we have demonstrated that hard-threshold is communication-optimal in our communication-complexity model. Our communication-complexity model is motivated by the EF-SGD non-convex convergence result, and it provides us insight into why hard-threshold has better convergence than Top-$k$ in practice.

---

> > ### Comment · Reviewer_TZ2h · 2021-08-31
> > **Comments after rebuttal**
> >
> > Many thanks to the authors' detailed replies and other reviewers' insights into this paper. I do appreciate the great value of this work, but I am still worried about the claim of optimality, even though I understand that this optimality is restricted to the model the authors proposed. My understanding is that from the upper bounds in Theorem 1, no matter what optimization trajectories, the upper bound only depends on the total compression errors, and the hard-threshod is optimal for any fixed optimization trajectories as long as the right threshold is chosen, and thus the optimality (please let me know if I misunderstood this). However, Theorem 1 only provides an upper bound, if we are minimizing an upper bound, claiming optimal seems to me kind of strong. I understand this upper bound motivates the hard-threshold compression methods, and the paper presents very solid theoretical and experimental results for this method. However, in terms of communication efficiency, i.e. bits over guaranteed optimization errors, there is no clear theoretical improvements shown, so maybe this can be some future works. I agree with other reviewers' votes and raise my score to 7.

---

> > > ### Author Response · Authors · 2021-09-01
> > > **Thank you for your positive reassessment**
> > >
> > > We thank the reviewer for the positive feedback, and for pointing out a meaningful direction for future research. We will clearly elaborate our use of the word optimal, as mentioned in the response to reviewer o9PJ.

---

### Decision · Program_Chairs · 2021-09-27

**Decision:**

Accept (Spotlight)

**Comment:**

This paper reformulates an existing problem (how to sparsify gradients in distributed training) and proposes to minimize a new objective (the total compression error subject to communication constraint as opposed to per-iteration compression error). This change of viewpoint leads to a new algorithm (hard threshold algorithm with variable sparsity). The authors show the effectiveness of the proposed algorithm through theoretical bounds and experiments. All reviewers agree that this is a valuable contribution. Comments from previous submission to ICML are adequately addressed.